# Sparsity-Preserving Differentially Private Training of Large Embedding Models

**Badih Ghazi**
Google Research
Mountain View, CA
badihghazi@gmail.com

**Yangsibo Huang**[*]
Princeton University
Princeton, NJ
yangsibo@princeton.edu

**Pritish Kamath**
Google Research
Mountain View, CA
pritishk@google.com

**Ravi Kumar**
Google Research
Mountain View, CA
ravi.k53@gmail.com

**Pasin Manurangsi**
Google Research
Mountain View, CA
pasin@google.com

**Amer Sinha**
Google Research
Mountain View, CA
amersinha@google.com

**Chiyuan Zhang**
Google Research
Mountain View, CA
chiyuan@google.com

## Abstract

As the use of large embedding models in recommendation systems and language applications increases, concerns over user data privacy have also risen. DP-SGD, a training algorithm that combines differential privacy with stochastic gradient descent, has been the workhorse in protecting user privacy without compromising model accuracy by much. However, applying DP-SGD naively to embedding models can destroy gradient sparsity, leading to reduced training efficiency. To address this issue, we present two new algorithms, DP-FEST and DP-AdaFEST, that preserve gradient sparsity during private training of large embedding models. Our algorithms achieve substantial reductions ($10^6\times$) in gradient size, while maintaining comparable levels of accuracy, on benchmark real-world datasets.

## 1 Introduction

Large embedding models have emerged as a fundamental tool for various applications in recommendation systems [CKH$^+$16, WFFW17, GTY$^+$17, ZZS$^+$18, NMS$^+$19, WSC$^+$21] and natural language processing [MSC$^+$13, PSM14, DCLT19]. Those models map categorical or string-valued input attributes with large vocabularies to fixed-length vector representations using embedding layers, which enable integrating non-numerical data into deep learning models. These models are widely deployed in personalized recommendation systems and achieve state-of-the-art performance in language tasks such as language modeling, sentiment analysis, and question-answering.

In the meantime, the use of large embedding models raises significant concerns about privacy violations, as it often entails the processing of sensitive individual information such as personally identifiable information, browsing habits, and dialogues [CTW$^+$21, HSC22, GHZ$^+$22]. Various techniques have been proposed to enable private data analysis. Among those, differential privacy (DP) [DMNS06, DKM$^+$06] is a widely adopted notion that could formally bound the privacy leakage of individual user information while still allowing for the analysis of population-level patterns. For training deep neural networks with DP guarantees, the most widely used algorithm is DP-SGD [ACG$^+$16]. It works by clipping the per-example gradient contribution, and noising the average gradient updates during each iteration of stochastic gradient descent (SGD). DP-SGD has

---

*This work was partially done while the author was interning at Google Research.

37th Conference on Neural Information Processing Systems (NeurIPS 2023).

demonstrated its effectiveness in protecting user privacy while maintaining model utility in a variety of applications [PTS⁺21, LTLH21, DFAP21, TB21, KCS⁺22, YNB⁺22, DBH⁺22, DGK⁺23].

However, applying DP-SGD to large embedding models presents unique technical challenges. These models typically contain non-numerical feature fields like user/product IDs and categories, as well as words/tokens that are transformed into dense vectors through an embedding layer. Due to the large vocabulary sizes of those features, the process requires embedding tables with a substantial number of parameters. In contrast to the number of parameters, the gradient updates are usually extremely sparse because each mini-batch of examples only activates a tiny fraction of embedding rows. This sparsity is heavily leveraged for industrial applications [GGDS19, WWL⁺22] that efficiently handle the training of large-scale embeddings. For example, Google TPUs [AF22], a family of accelerators designed specifically for training large-scale deep networks, have dedicated hardware such as the SparseCore [JKL⁺23] to handle large embeddings with sparse updates. It is shown [Sna22] that this leads to significantly improved training throughput compared to training on GPUs, which did not have specialized optimization for sparse embedding lookups at the time. On the other hand, DP-SGD completely destroys the gradient sparsity as it requires adding independent Gaussian noise to *all* the coordinates. This creates a road block for private training of large embedding models as the training efficiency would be significantly reduced compared to non-private training.

**Our contributions.**    In this paper, we present new algorithms that target the issue of diminished gradient sparsity encountered when employing DP-SGD with large embedding models due to the addition of dense noise; this issue represents a significant practical obstacle for leveraging, during DP training, the hardware accelerators commonly used in non-private training of large embedding models. While there are previous studies of the abstract problems of sparse vectors in DP, we study the practical problem of privately training large embedding models, and through systematic evaluations, demonstrate that significant gradient size reduction is achievable in DP training of real-world large-scale embedding models (e.g., recommender systems, language models with $> 30M$ embedding parameters) with minimal loss in utility. Our contributions can be summarized as follows:

▷ We propose two new algorithms to preserve gradient sparsity in DP-training of large embedding models while maintaining their utility (Section 3). The first algorithm, **F**iltering-**E**nabled **S**parse **T**raining (DP-FEST), selectively adds noise to pre-selected embedding rows during training. To improve the adaptivity to varying training dynamics across mini-batches and to accommodate online training, our second algorithm, **Ada**ptive **F**iltering-**E**nabled **S**parse **T**raining (DP-AdaFEST), adaptively computes a gradient contribution map for each mini-batch, and only injects noise into the gradients of top contributors (in a manner that complies with the DP requirements).

▷ We demonstrate the effectiveness of our algorithms on four benchmark datasets for online advertising and natural language understanding. Our results indicate that these algorithms can meet a wider range of application-specific demands for utility and efficiency trade-offs at a more granular level (Section 4.2). Notably, they achieve a substantially sparser gradient, with a reduction in gradient size of over $10^6 \times$ compared to the dense gradient produced by vanilla DP-SGD, while maintaining comparable levels of accuracy.

▷ We further showcase the potential applicability and effectiveness of our algorithms to the setting of online learning with streaming data, a common setup in practice, which highlights the versatility and broad impact of our algorithms (Section 4.3).

## 2   Preliminaries

### 2.1   Embedding Layers and Gradient Sparsity

Models with large embedding layers are widely used in application scenarios that need to process high-dimensional sparse inputs such as words/tokens in language models [MSC⁺13, PSM14, DCLT19] and categorical features associated with users or items in recommendation systems [CKH⁺16, WFFW17, GTY⁺17, ZZS⁺18, NMS⁺19, WSC⁺21].

Let $c$ be the number of possible values of an input feature (aka *vocabulary size* in text models); the feature values are also called *feature buckets* (string-valued categorical features are commonly preprocessed via hashmap bucketization). An input $\mathbf{x}$ with the $i$th feature value can be represented as a one-hot vector $\mathbf{e}_i \in \mathbb{R}^c$ that is 1 on the $i$th coordinate and 0 everywhere else. Let $\mathbf{W} \in \mathbb{R}^{c \times d}$ be the parameters of the embedding layer (i.e., the embedding table), where the output dimension $d$ is the

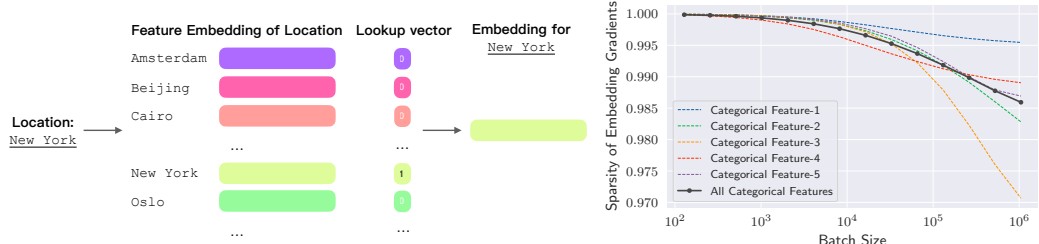

(a) An illustration of embedding lookup.     (b) Embedding gradient sparsity in Ads model.

Figure 1: Illustration of embedding lookup (a) and gradient sparsity of the Criteo pCTR model (b). We analyze the gradient sparsity, averaged over 50 update steps, of the top five categorical features (out of a total of 26) with the highest number of buckets, as well as the sparsity of all categorical features. For more information about this dataset and model, see Section 4.1.1.

*embedding size*. The output of the embedding layer is computed as $\mathbf{z} = \mathbf{W}^\top \mathbf{x}$, which is a linear map. Due to efficiency considerations, in practice, embedding layers are rarely implemented via matrix multiplication (`matmul`) on one-hot encoded inputs.

For a one-hot input $\mathbf{x} = \mathbf{e}_i$, the embedding output is the $i$th row of the embedding table: $\mathbf{z} = \mathbf{W}[i, :]$. Figure 1a illustrates the embedding table lookup operation. Let $\partial \mathcal{L} / \partial \mathbf{z} \in \mathbb{R}^d$ be the partial derivative of the loss with respect to the embedding output. The gradient of the embedding table is $\nabla W = \mathbf{x} \otimes \partial \mathcal{L} / \partial \mathbf{z}$, where $\otimes$ is the outer product. For a one-hot input $\mathbf{x} = \mathbf{e}_i$, the result of this outer product is a sparse matrix whose $i$th row equals $\partial \mathcal{L} / \partial \mathbf{z}$. For mini-batch SGD training, the number of non-zero rows in the batch averaged gradient of the embedding table is upper bounded by the batch size, which is typically orders of magnitude smaller than $c$. Consequently, the *gradient sparsity* (the fraction of zero gradient coordinates) for large embedding models is very high (Figure 1b).

Due to this structured sparsity, in practice, both forward and backward computations of an embedding layer are implemented efficiently with gathers and scatters[1], without expensive `matmul`. This difference from vanilla linear layers is crucial in real-world applications as the embedding tables are usually very large, with the vocabulary size $c$ ranging from tens of thousands (language models) to millions (recommendation systems). Moreover, in some large recommendation system models, there are hundreds of different categorical features, each with a different embedding table. Therefore, maintaining the gradient sparsity is critical for any training process.

For notational simplicity, the description above focuses on single-variate features, which only activate one feature value at a time. In practice, multi-variate features activating multiple values are also used. In this case, it is common for the embedding layer to output the average or sum vectors corresponding to each activated value. We also note that in recommendation systems, each categorical feature generally has a separate embedding table; But for text models, all the words/tokens in an input document share the same embedding table.

## 2.2 Differentially Private Stochastic Gradient Descent

*Differential privacy (DP)* [DMNS06, DKM⁺06] is a mathematical framework for ensuring the privacy of individuals in datasets. It can provide a strong guarantee of privacy by allowing data to be analyzed without revealing sensitive information about any individual in the dataset. Formally, a randomized algorithm $\mathcal{A}$ satisfies $(\varepsilon, \delta)$-*DP* if for any two neighboring datasets $\mathcal{D}$ and $\mathcal{D}'$ (i.e., datasets such that one can be obtained from the other by adding/removing one example), and any subset $\mathcal{S}$ of outputs, it holds for privacy parameters $\varepsilon \in \mathbb{R}_{>0}$ and $\delta \in [0, 1)$ that

$$\Pr[\mathcal{A}(\mathcal{D}) \in \mathcal{S}] \leq e^\varepsilon \cdot \Pr[\mathcal{A}(\mathcal{D}') \in \mathcal{S}] + \delta.$$

*DP Stochastic Gradient Descent* (`DP-SGD`) [ACG⁺16] is a recipe for training a deep learning model with DP by modifying the mini-batch stochastic optimization process through the use of per-example gradient clipping and Gaussian noise injection. When training an ML model $f$ parameterized by $\theta$ with the per-example loss function $\ell(\cdot, \cdot)$[2] on dataset $\mathcal{D}$, each optimization step $t$ involves randomly

---

[1]Gather/scatter refers to a memory addressing technique that enables simultaneous collection (gathering) from or storage (scattering) of data to multiple arbitrary indices. Refer to this wiki page for more details.

[2]The specific loss depends on the particular task and model (e.g. cross-entropy loss for classification).

sampling a mini-batch $\mathcal{B}_t$. Given $\mathcal{B}_t$, DP-SGD starts by computing the per-example gradient for each $(x_i, y_i) \in \mathcal{B}_t$, where $x_i$ is the feature vector and $y_i$ is the corresponding label, as follows:

$$\mathbf{g}_t(x_i, y_i) \leftarrow \nabla_{\theta_t} \ell\left(f_{\theta_t}(x_i), y_i\right).$$

It then *clips* the gradient $\ell_2$-norm to a maximum $\ell_2$-norm of $C$ as:

$$[\mathbf{g}_t(x_i, y_i)]_C := \mathbf{g}_t(x_i, y_i) \Big/ \max\left(1, \frac{\|\mathbf{g}_t(x_i, y_i)\|_2}{C}\right).$$

Finally, it produces the private gradient $\hat{\mathbf{g}}_t$ by injecting Gaussian noise into the sum of the clipped per-example gradients as:

$$\hat{\mathbf{g}}_t \leftarrow \frac{1}{\|\mathcal{B}_t\|}\left(\sum_i [\mathbf{g}_t(x_i, y_i)]_C + \mathcal{N}\left(0, \sigma^2 C^2 \mathbf{I}\right)\right), \tag{1}$$

where $\mathcal{N}(0, \sigma^2 C^2 \mathbf{I})$ is a Gaussian distribution with mean 0 and covariance $\sigma^2 C^2 \mathbf{I}$, and the noise multiplier $\sigma$ is computed from $(\varepsilon, \delta)$ by inverse privacy accounting (e.g., [ACG$^+$16]).

Although DP-SGD has been demonstrated to be an effective algorithm for training ML models with DP, (1) requires adding noise to *all* the coordinates of the gradient, which would completely destroy any existing sparsity structure in the original gradients. This densification of sparse gradients is especially problematic for large industrial-scale embedding models, where the sparsity is heavily leveraged to improve efficiency, e.g., using dedicated APIs [GGDS19, AF22, WWL$^+$22, Sna22].

## 3 Sparsity-Preserving DP-SGD

In this section, we introduce two algorithms for preserving the gradient sparsity of DP-SGD when training large embedding models. Our main intuition is that in real-world datasets, some buckets of categorical features may be much more frequent, and therefore contain more significant or relevant information, than others. However, vanilla DP-SGD adds noise to the model's parameter gradients, even to those with minimal impact on the model's accuracy, which is not only wasteful but also can reduce the training efficiency by disrupting gradient sparsity (as discussed earlier in Section 2.2).

### 3.1 Filtering-Enabled Sparse Training (DP-FEST)

A simple solution to this problem is *frequency filtering*, where we pre-select the top-$k$ most informative (i.e., frequent) buckets of each categorical feature before training, and add Gaussian noise only to the gradients of these selected buckets during training; this is akin to training a smaller embedding model using a subset of the buckets. Frequency filtering significantly reduces the noise added to the gradients by restricting the noise to only the most impactful subset of features.

There are two possible ways of selecting the top-$k$ buckets. First, if there is prior information on bucket frequency available publicly (e.g., token frequency in the pre-training set of the model for language tasks), then it can be used to select the top-$k$ buckets. If such prior knowledge is unavailable, we can run DP top-$k$ selection on the training dataset by adapting the algorithm of [DR21]; see Appendix B.1 for details.

### 3.2 Adaptive Filtering-Enabled Sparse Training (DP-AdaFEST)

The limitation of frequency filtering in DP-FEST is its inability to adapt to varying activation patterns across different mini-batches during training. This issue becomes particularly problematic when dealing with streaming data, such as those used in recommendation systems. In such scenarios, the application of frequency filtering poses a challenge as it impedes training until a substantial amount of data is gathered to accurately estimate the frequency information.

To address this issue, we propose DP-AdaFEST, an adaptive method for selecting the most informative buckets using mini-batch information, as outlined in Figure 2 and Algorithm 1. For each mini-batch, DP-AdaFEST first computes the per-example gradient and a gradient contribution map, which is a binary vector indicating which categorical feature buckets have been activated for each example in the mini-batch (Line 5). These per-example gradient contributions are then aggregated (with per-example clipping applied) across the mini-batch in order to construct the batch-wise gradient contribution,

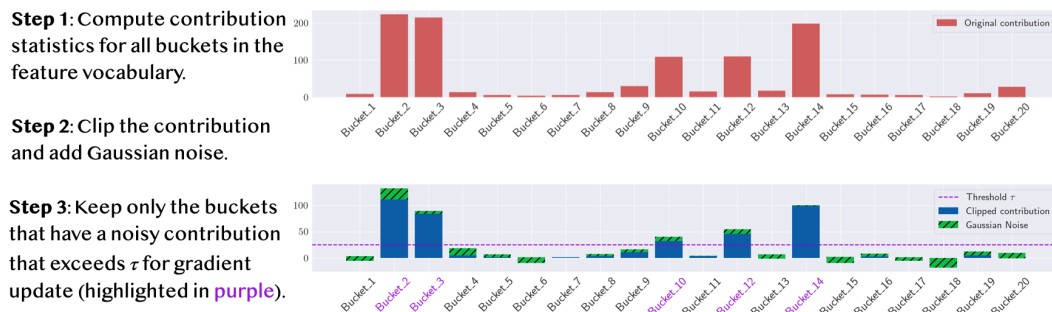

**Step 1:** Compute contribution statistics for all buckets in the feature vocabulary.

**Step 2:** Clip the contribution and add Gaussian noise.

**Step 3:** Keep only the buckets that have a noisy contribution that exceeds $\tau$ for gradient update (highlighted in purple).

Figure 2: Illustration of the process of DP-AdaFEST on a synthetic categorical feature which has 20 buckets. We compute the number of examples contributing to each bucket, adjust the value based on per-example total contributions (including those to other features), add Gaussian noise, and retain only those buckets with a noisy contribution exceeding the threshold for (noisy) gradient update.

---

**Algorithm 1:** DP-AdaFEST: Adaptive Filtering Applied Sparse Training.

**Input** : Learning rate $\eta$, batch size $B$, number of training steps $T$, dataset $\mathcal{D}$, clipping norms $C_1$, $C_2$, threshold parameter $\tau$, noise multipliers $\sigma_1$, $\sigma_2$.

**Output :** DP-trained model parameters $\theta$.

1 Initialize model parameters $\theta$ randomly;
2 **for** $t = 1$ **to** $T$ **do**
3      Sample a mini-batch $\mathcal{B}_t$ of size $B$ from $\mathcal{D}$;
4      **for** $i = 1$ **to** $B$ **do**
5          Compute per-example gradient $g_i$ and gradient contribution map $v_i$, where $v_i[j] := \mathbf{1}[g_i[j,:] \neq \mathbf{0}]$;
6      Compute the private contribution map $V_t = \sum_{i \in \mathcal{B}_t} [v_i]_{C_1} + C_1 \cdot \mathcal{N}\left(0, \sigma_1^2 \cdot I_c\right)$;
7      **for** $i = 1$ **to** $B$ **do**
8          $g_i[j,:] \leftarrow \mathbf{0}$, for all $j$ such that $V_t[j] < \tau$ ;
9      $G_t = \sum_{i \in \mathcal{B}_t} [g_i]_{C_2} + C_2 \cdot \mathcal{N}\left(0, \sigma_2^2 \cdot \text{diag}\left(\left(1_{\{V_t[j] \geq \tau\}}\right)_{j \in [c]} \otimes 1_d\right)\right)$;
10      Update the model parameters: $\theta \leftarrow \theta - \eta G_t$;

---

and Gaussian noise with scale $\sigma_1$ is added to the resulting vector to ensure DP (Line 6).[3] The noisy batch-wise gradient contribution is then thresholded using a parameter $\tau$ to exclude non-significant gradient entries to which only a few examples in the mini-batch contribute (Line 8); This thresholding mechanism helps focus on the most relevant and informative features while reducing the impact of noisy or insignificant contributions. Finally, the algorithm updates the model parameters by adding gradient noise with scale $\sigma_2$ only to the "surviving" gradient entries (Lines 9 and 10). By dynamically adapting feature selection based on bucket contributions in each mini-batch during training, our algorithm achieves adaptive feature selection and maintains the desired DP guarantees. Note that we apply the standard DP-SGD with noise multiplier $\sigma_2$ to update parameters in non-embedding layers.

### 3.3 Privacy Accounting

The privacy accounting of DP-AdaFEST can be done in a similar way to that of DP-SGD. The main observation is that the privacy cost of Algorithm 1 is dictated by the Gaussian noise mechanism steps in Lines 6 and 9. In particular, the privacy cost of a single iteration is equivalent to that of the composition of two Gaussian mechanisms with noise scale $\sigma_1$ and $\sigma_2$ respectively, which in turn is equivalent to the privacy cost of a single Gaussian mechanism of noise scale $\sigma = \left(\sigma_1^{-2} + \sigma_2^{-2}\right)^{-1/2}$. Thus, overall, the privacy cost of the entire algorithm is no more than that of DP-SGD with noise scale $\sigma$, and identical other parameters of batch size and number of steps. Finally, we adopt the standard approach of analyzing the Poisson subsampled Gaussian mechanism [ACG+16], using numerical algorithms based on privacy loss distributions [KJH20, GLW21, GKKM22, DGK+22]. In particular, we use the open-source implementation available in Google's DP library [Goo20]. We provide more details in Appendix C for completeness.

---

[3]When implemented naively, this step has a memory consumption that scales with the number of embedding rows. In Appendix B.2, we discuss an alternative method for sampling the surviving coordinates of the mask in a memory-efficient manner.

An important caveat to the privacy analysis is that we follow a common practice of analyzing all the algorithms where the mini-batches are *Poisson subsampled*, wherein each example is included independently with a certain probability, even though, the actual implementation involves selecting mini-batches of fixed sizes after shuffling. See Section 4.3 of [PHK+23] for more context. But since this caveat applies to all algorithms similarly, we consider their relative comparison to be fair.

### 3.4 Bias-Variance Trade-offs

Apart from the computational advantage, to understand the question of when DP-AdaFEST would lead to more accurate models as compared to DP-SGD, we look through the lens of bias-variance trade-offs in stochastic convex optimization. Consider a *convex* objective $\mathcal{L}(\cdot)$ over $\mathbb{R}^D$, for which we have a gradient oracle, that given $\theta$, returns a stochastic estimate $g(\theta)$ of $\nabla \mathcal{L}(\theta)$. We say that the gradient oracle has bias $\alpha$ and variance $\sigma^2$ if $g(\theta) = \nabla \mathcal{L}(\theta) + \zeta(\theta)$ such that $\|\mathbb{E}\,\zeta(\theta)\|_2 \leq \alpha$ and $\mathbb{E}\,\|\zeta(\theta) - \mathbb{E}\,\zeta(\theta)\|_2^2 \leq \sigma^2$ holds for all $\theta$. When optimizing over a convex set $\mathcal{K} \subseteq \mathbb{R}^D$, projected gradient descent with step size $\eta$ is defined as iteratively performing $\theta_{t+1} \leftarrow \Pi_{\mathcal{K}}(\theta_t - \eta g(\theta_t))$, where $\Pi_{\mathcal{K}}(\cdot)$ is the projection onto $\mathcal{K}$. We recall the following guarantee on the expected excess loss obtained using standard analysis of projected stochastic gradient descent (see e.g. [Haz22]).

**Lemma 3.1.** *For an $L$-Lipschitz loss function, and a gradient oracle with bias $\alpha$ and variance $\sigma^2$, projected gradient descent over a set $\mathcal{K}$ with diameter $R$, with step size $\eta = \frac{R}{\sqrt{((L+\alpha)^2 + \sigma^2)T}}$ achieves*

$$\mathbb{E}\left[\mathcal{L}\left(\frac{1}{T}\sum_{i=1}^{T}\theta_i\right)\right] - \mathcal{L}(\theta^*) \; \leq \; \frac{R}{\sqrt{T}}\cdot\sqrt{(L+\alpha)^2 + \sigma^2} + \alpha R.$$

Suppose the loss is $L$-Lipschitz. Ignoring the potential bias introduced due to clipping, DP-SGD of noise scale $\sigma$ uses a gradient oracle with zero bias ($\alpha = 0$) and variance $D\sigma^2$. On the other hand, consider a hypothetical setting where DP-AdaFEST truncates $\gamma$ fraction of the gradient due to masking in Line 8, resulting in the final gradient that is supported on $h$ coordinates ($h \ll D$). In this case, the bias introduced is $\gamma L$, but the variance introduced is only $h\sigma^2$. Plugging into the above excess risk bound, we would expect DP-AdaFEST to achieve a smaller excess loss when

$$\sqrt{L^2(1+\gamma)^2 + h\sigma^2} + \gamma L \sqrt{T} \; < \; \sqrt{L^2 + D\sigma^2}\,. \tag{2}$$

This implies that a smaller truncated fraction within DP-AdaFEST could potentially yield superior utility compared to vanilla DP-SGD. Even though the discussion above is for the average iterate, and in the convex setting, our experiments corroborate this observation in training deep learning models.

## 4 Experiments

In this section we evaluate the performance of our sparsity-preserving training algorithms and compare them against vanilla DP-SGD on both the recommendation and language understanding tasks (Section 4.1). To understand adaptivity, we evaluate our algorithms on a time-series version of the data (Section 4.3); the non-time-series evaluations are in Section 4.2. We further demonstrate the applicability of our methods on language models in Section 4.4. The impact of hyper-parameters on the trade-off between utility and embedding gradient size is discussed in Section 4.5.

### 4.1 Setup

#### 4.1.1 Datasets and Models

**Click-through rate prediction tasks.** We evaluate our algorithms on the widely-used Criteo predicted click-through rate (pCTR) dataset[4], which includes over four billion ad impressions over 24 days. Each impression is represented by 13 numerical features and 26 categorical features. The objective is to predict how likely a user clicks on an ad based on these features.

The pCTR model we experiment with is a neural network that uses embedding layers for categorical features and log transformations for numerical features, followed by several fully connected layers. We use binary cross-entropy loss as the training objective and report the AUC as the evaluation metric. For further details about the training setup, please refer to Appendix D.1.

We evaluate two Criteo variants with different purposes:

---

[4] https://ailab.criteo.com/download-criteo-1tb-click-logs-dataset

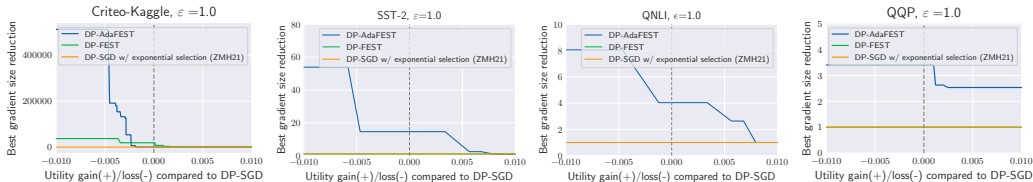

Figure 3: A comparison of the best gradient size reduction achieved by DP-AdaFEST, DP-FEST, and DP-SGD with exponential selection [ZMH21] compared to DP-SGD at different thresholds for utility difference. A higher curve indicates a better utility/efficiency trade-off. DP-AdaFEST consistently outperforms DP-FEST.

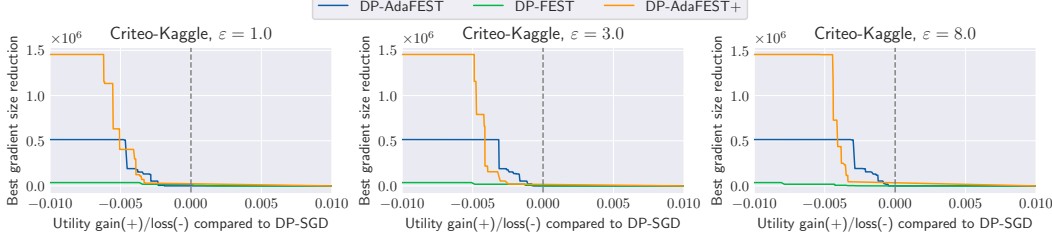

Figure 4: A comparison between DP-AdaFEST, DP-FEST, and the combined algorithm (DP-AdaFEST+) on the Criteo-Kaggle dataset for different values of $\varepsilon$. The combined algorithm significantly outperforms either alone. Figure 6 demonstrates similar findings on the Criteo-time-series dataset.

▷ *Criteo-Kaggle*[5], a subset of examples released by Kaggle (~45 million examples in total), is widely used in previous studies of click-through rate modeling. We use this to benchmark the performance of our algorithms. Note that timestamps are absent in Criteo-Kaggle.

▷ *Criteo-time-series*, also commonly known as "Criteo-1TB", is the entire Criteo dataset of over 4 billion examples and has the auxiliary information indicating which day the data was collected. To simulate a real-world online training scenario, we train the model on the first 18 days of data and evaluate it on subsequent days (i.e., days 19–24); more details on the training process are in Section 4.3.

**Language understanding tasks.** For language understanding tasks, we employ language models from the BERT family [DCLT19], specifically the RoBERTa model [LOG+19]. This model has a vocabulary size of $50,265$ subword tokens and has been pre-trained on public web data.

We fine-tune the RoBERTa model for downstream classification tasks from the GLUE benchmark [WSM+19], including SST-2 [SPW+13], QNLI [RZLL16], and QQP [IYW+17]. Following [YNB+22], we adopt Low-Rank Adaptation (LoRA) [HSW+22] to introduce trainable rank decomposition matrices into each transformer block of the language model. This approach significantly reduces the number of parameters required for downstream tasks while also improving the privacy-utility trade-off (see Appendix D.1). A notable difference in our approach compared to [YNB+22] is that we also train the word embedding layers in DP fine-tuning to enable bucket selection. This leads to significant accuracy improvements in the model, as shown in Table 6.

### 4.1.2 Baseline and Algorithms

We evaluate the performance of vanilla DP-SGD, and the three sparsity-preserving algorithmic variants, DP-SGD with exponential selection [ZMH21], DP-FEST and DP-AdaFEST, on both the click-through rate prediction and language understanding tasks. We use a fixed batch size of $2,048$ for the former and $1,024$ for the latter. For all tasks, we set the privacy parameter $\delta$ to $1/N$, where $N$ is the number of training examples for the respective task.

### 4.2 Evaluation on Non-Time-Series Data

We begin by considering datasets with no timestamp—Criteo-Kaggle, SST-2, QNLI, and QQP.

The decision to prioritize gradient sparsity (a key factor impacting efficiency) or utility in DP training usually depends on the task and available computational resources. However, it can be challenging for vanilla DP-SGD to cater to different needs of gradient sparsity or utility since it lacks mechanisms for trading off between these objectives. In contrast, DP-SGD with exponential selection [ZMH21],

---

[5]http://labs.criteo.com/2014/02/kaggle-display-advertising-challenge-dataset

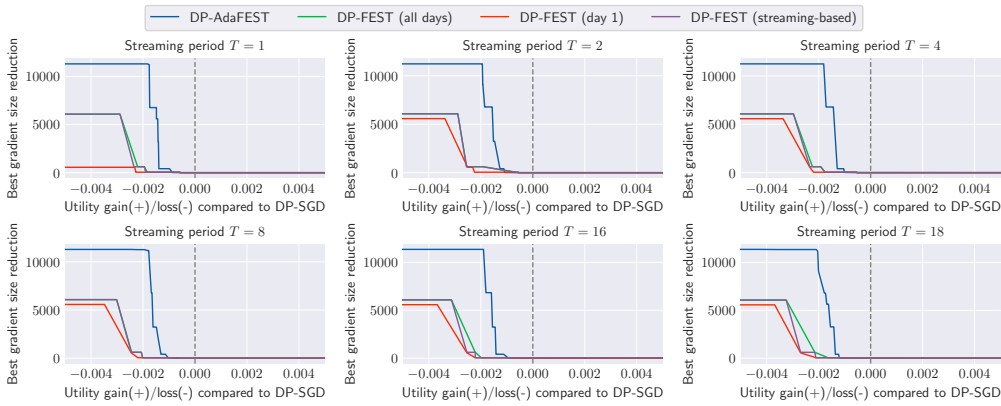

Figure 5: A comparison between DP-AdaFEST and DP-FEST for time-series data, using the Criteo-time-series dataset with different streaming periods $T$ and $\varepsilon = 1.0$. DP-AdaFEST consistently achieves a higher gradient reduction than DP-FEST at the same level of utility.

DP-AdaFEST and DP-FEST can all offer diverse options for balancing utility and efficiency via the sparsity-controlling parameters (see Figure 8 in Appendix D.2), while our proposals DP-AdaFEST and DP-FEST offer much better privacy-utility loss.

While both DP-FEST and DP-AdaFEST offer ways to balance efficiency and utility in DP training, DP-AdaFEST stands out as the *more versatile and customizable* algorithm. With three adjustable hyper-parameters, DP-AdaFEST offers more diverse results compared to DP-FEST's single knob, $k$, which controls the number of preserved top buckets. The effectiveness of DP-AdaFEST is evident in Figure 3, where it achieves significantly higher gradient size reduction than DP-FEST while maintaining the same level of utility. Specifically, on the Criteo-Kaggle dataset, DP-AdaFEST reduces the gradient computation cost of vanilla DP-SGD by more than $5 \times 10^5$ times while maintaining a comparable AUC (of AUC loss less than 0.005). This reduction translates into a more efficient and cost-effective training process (Appendix D.2.1 shows wall-clock time improvements). Although the best reduction for language tasks is usually smaller since the vocabulary is already relatively condensed, the adoption of sparsity-preserving DP-SGD effectively obviates the dense gradient computation. Furthermore, in line with the bias-variance trade-off presented in Section 3.4, we note that DP-AdaFEST occasionally exhibits superior utility compared to DP-SGD when the reduction in gradient size is minimal (i.e., $\gamma$ in Equation (2) is small). Conversely, when incorporating sparsity, DP-SGD with the exponential mechanism faces challenges in maintaining utility. In all configurations, it fails to achieve a tolerable level of utility loss.

We further explore the potential of integrating DP-AdaFEST with DP-FEST to enhance performance, which involves pre-selecting a subset of buckets using DP-FEST and subsequently training on this subset using DP-AdaFEST. Figure 4 shows that the combined algorithm (DP-AdaFEST+) outperforms either alone in utility/efficiency trade-off, with the best gradient size reduction being further improved to $> 10^6 \times$. This can be attributed to the complementary strengths of the two algorithms: DP-AdaFEST offers a more flexible approach to feature selection at the batch level, while DP-FEST provides a simple yet effective means for feature selection according to the global frequency information. Besides, the combined algorithm expands the range of choices for balancing gradient sparsity and utility through the combination of hyper-parameters from both DP-FEST and DP-AdaFEST.

## 4.3 Evaluation on Time-Series Data

Time-series data are notoriously challenging due to non-stationarity. In this section, we investigate if our algorithms can adapt to distribution shifts, characteristic of time-series data.

We conduct experiments using the Criteo-time-series dataset, which comprises real-world user-click data collected over 24 days (18 days for training and the rest for evaluation). To simulate the online data streaming scenario, we introduce the concept of a *streaming period*, a time interval during which the model is updated as new data is received. Experiments indicate a notable disparity between DP training (using vanilla DP-SGD) and non-DP training, with the former demonstrating increased susceptibility to distribution shifts (Table 5 in Appendix D.2).

Figure 5 presents the utility and efficiency trade-off of DP-AdaFEST and DP-FEST for time-series data. To explore the efficacy of DP-FEST, we investigate different sources of vocabulary frequency, including the information from the first day, all days, and streaming-based (i.e., running sum updated

per streaming period). The experimental results indicate that using streaming-based frequency information for DP-FEST is almost as effective as using all-day frequency information and significantly better than using information only from the first day. Additionally, DP-AdaFEST consistently outperforms DP-FEST, achieving more than twice the gradient reduction at the same level of utility. These findings further demonstrate the importance of adapting to the dynamic nature of time-series data at a more fine-grained level, with DP-AdaFEST able to adapt at the batch level and DP-FEST only able to adapt at the streaming period level.

We further examine the advantages of combining DP-AdaFEST and DP-FEST to enhance the utility/efficiency trade-off on the Criteo-time-series dataset. We use a streaming period of 1 and streaming-based frequency information for DP-FEST. Figure 6 demonstrates a consistent superiority of the combined approach over individual methods, which aligns with the findings presented in Section 4.3.

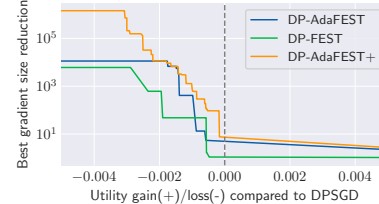

Figure 6: The combined approach (DP-AdaFEST+) outperforms either alone on Criteo-time-series.

## 4.4 Applicability to Language Models

We further demonstrate the applicability of our method to language models by comparing it with LoRA [HSW⁺22] and reporting its performance with multilingual models.

**DP-AdaFEST is a better choice than LoRA for embedding layers.** LoRA was introduced as a method to efficiently adapt matrices of dimensions $n \times d$ in language models by utilizing a rank-$r$ approximation, where the initial use case considers the attention layers where $n = d$. The rank-$r$ approximation helps in reducing the memory requirements by a factor of $m \times d/(m + d) * r < \min(m, d)/r$. However, the applicability of LoRA to embedding layers is limited for several reasons:

▷ **Limited improvement for unbalanced $n$ and $d$:** Embedding layers typically involve a large vocabulary size ($n$ often in the millions) and a smaller embedding dimension ($d$ typically in the hundreds). In this context, LoRA's potential benefits are limited.

▷ **Inability to harness APIs**: For private training of the embedding layer, DP-AdaFEST allows for efficient embedding lookup operations (row fetching) using custom APIs; In contrast, LoRA still relies on computationally costly matrix multiplication and cannot effectively utilize these APIs.

▷ **Limited to fine-tuning use cases**: LoRA is designed for adapting pre-trained models, while our DP-AdaFEST is versatile, functioning seamlessly in both pre-training and fine-tuning scenarios.

To more effectively demonstrate the first argument above, Table 1 compares the best embedding gradient size reductions achieved by DP-AdaFEST and LoRA against DP-SGD for SST-2 with $\varepsilon = 1.0$, on the RoBERTa model. We vary LoRA's rank $r$ from $\{4, 8, 16, 32, 64, 128\}$. DP-AdaFEST consistently outperforms LoRA in gradient size reduction at similar utility levels.

| Utility loss compared to DP-SGD | Gradient size reduction | |
|---|---|---|
| | DP-AdaFEST | LoRA |
| 0.001 | 14.59× | 5.91× |
| 0.005 | 53.90× | 23.64× |
| 0.01 | 53.90× | 47.28× |

Table 1: Gradient size reduction by LoRA and DP-AdaFEST for RoBERTa's word embeddings.

**Increased gradient size reduction by DP-AdaFEST for larger vocabularies.** Our evaluation in Section 4.2 primarily centered around the RoBERTa model, which has a vocabulary of around $50,000$ entries. However, it is important to note that many high-vocabulary models exist, with vocabulary sizes 5 to $20\times$ larger than RoBERTa. For these models, DP-AdaFEST could offer even more pronounced benefits.

| Utility loss compared to DP-SGD | Gradient size reduction | |
|---|---|---|
| | RoBERTa ($|V|$:50k) | XLM-R ($|V|$:250k) |
| 0.001 | 14.59× | 17.21× |
| 0.005 | 53.90× | 64.75× |
| 0.01 | 53.90× | 152.94× |

Table 2: DP-AdaFEST offers more pronounced gradient size reduction for models with larger vocabulary (i.e., higher $|V|$). RoBERTa results are achieved on SST-2 with $\varepsilon = 1.0$, and the XLM-R [CKG⁺20] results are achieved on the Cross-Lingual Natural Language Inference (XNLI) dataset [CRL⁺18] with $\varepsilon = 1.0$.

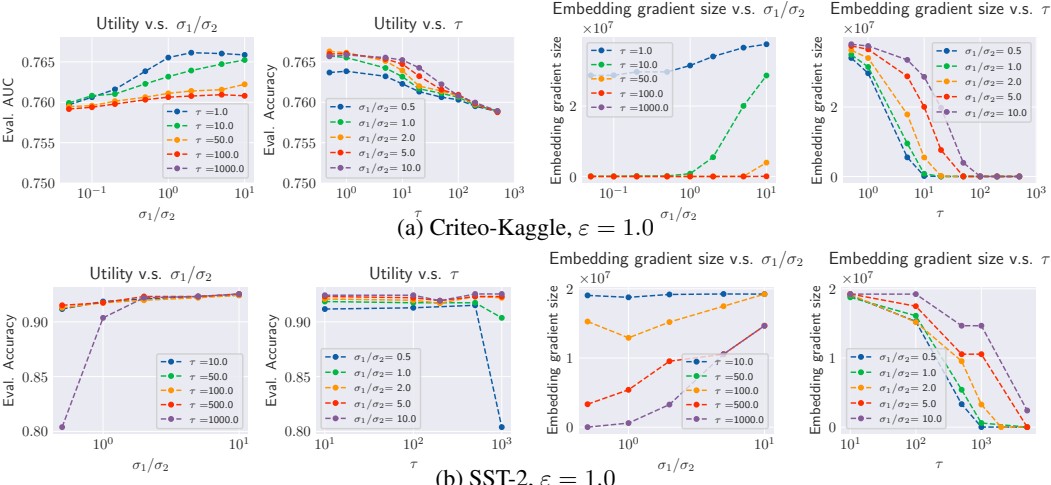

Figure 7: Effect of hyper-parameters on utility and embedding gradient size, including the ratio of noise added to the contribution map to the one added to the sparse gradient $\sigma_1/\sigma_2$, and the thresholding value $\tau$. The joint impact of two hyper-parameters on utility and embedding gradient size can be found in Figure 9 in Appendix D.2.

## 4.5 Effect of Hyper-Parameters

Finally, we study the effect of hyper-parameters on the utility and gradient size trade-off in `DP-AdaFEST`. Specifically, we explore the effects of the ratio of noise added to the contribution map and the noise added to the sparse gradient (i.e., $\sigma_1/\sigma_2$), and the thresholding value $\tau$.

**Effect of $\sigma_1/\sigma_2$.** We find that a larger ratio of $\sigma_1/\sigma_2$ leads to a higher accuracy (Figure 7). Indeed, the contribution map can tolerate higher levels of noise, leading to a more accurate representation of the sparse gradient. For both Criteo-Kaggle and SST-2 datasets, a noise ratio of 5 or 10 delivers the best utility. On the other hand, a larger $\sigma_1/\sigma_2$ results in a higher gradient density because the contribution map can tolerate higher noise levels, resulting in more zero-contribution buckets bypassing the threshold and increasing false negatives. Therefore, choosing an appropriate $\sigma_1/\sigma_2$ is critical to achieving optimal model accuracy and gradient density.

**Effect of $\tau$.** The choice of the thresholding value, $\tau$, is also crucial for determining the balance between gradient sparsity and model accuracy in `DP-AdaFEST`. Figure 7 reveals that increasing the value of $\tau$ usually leads to a decrease in the gradient size, which results in a sparser gradient. We also find that when $\tau$ is small, increasing it does not significantly impact the model's performance. However, setting an excessively high value of $\tau$ (e.g., $\tau > 500$ for a batch size of 1024) can cause a sharp drop in the model's accuracy, especially when the noise ratio $\sigma_1/\sigma_2$ is small; In such cases, important contributions may be zeroed out, resulting in a loss of information in the gradient. Again, it is critical to choose $\tau$ to best balance gradient sparsity and model accuracy in `DP-AdaFEST`.

## 5 Conclusions

In this work we present new algorithms—`DP-FEST` and `DP-AdaFEST`—for preserving gradient sparsity in DP training, particularly in applications involving large embedding models. Our algorithms achieve significant reductions in gradient size while maintaining accuracy on real-world benchmark datasets. We also provide recommendations on hyper-parameter settings.

**Limitations and future work.** Despite the advancements made in our proposed algorithms, there are still limitations and areas for future work. One such area is the exploration of customized hardware acceleration for sparse DP training, which has the potential to significantly improve the training efficiency of our models. By leveraging specialized hardware, we can further optimize the computational performance and speed up the training process. Additionally, combining our algorithms with other privacy-preserving techniques (e.g., Federated Learning) could lead to new insights and applications.

## Acknowledgment

Yangsibo Huang is supported by Princeton University's Wallace Memorial Fellowship.

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

# A   Related Work

Embedding is a fundamental ML technique used in modern deep learning to represent non-numerical features as dense numerical vectors in a key-value store format. In particular, embedding layers play a crucial role in recommendation systems [CKH$^+$16, WFFW17, GTY$^+$17, ZZS$^+$18, NMS$^+$19, WSC$^+$21], with models relying heavily on them for effective predictions; they usually have a small number of parameters in their dense layers but their embedding layers can contain billions of entries and multiple billions of parameters. However, these embedding layers differ significantly from those used in other deep learning tasks, such as natural language processing [MSC$^+$13, PSM14, DCLT19], where models such as BERT [DCLT19] use fewer entries in their embedding layers but compensate with several hundred million parameters in their dense feed-forward and attention layers.

The exponential growth of data is leading to embedding tables reaching sizes spanning multiple gigabytes to terabytes, which presents unique training challenges. For instance, fitting Google's 1.2TB model [MNW$^+$22], Baidu's 10TB model [ZXJ$^+$20], and Meta's 12 TB model [MHH$^+$22] onto a single compute node or accelerator requires significant effort. Therefore, optimizing the forward (sparse lookup) and backward (gradient backpropagation) passes for embedding layers is critical to unlock the high compute throughput of embedding models, particularly for recommendation systems. Recent research has focused on designing optimized embedding layers [AF22, JKL$^+$23] and developing efficient sparse optimizers [GGDS19, WWL$^+$22]. Furthermore, data parallelism techniques, such as distributing embedding tables across multiple compute nodes, have been explored in order to improve performance [Sna22].

However, optimizing embedding models trained with DP is an unexplored area. A recent work [DGK$^+$23] demonstrates the effectiveness of vanilla `DP-SGD` in the context of recommendation tasks; however, this study does not address new challenges such as the need for efficient sparse gradient computation and adaptive noise addition to comply with privacy guarantees. Another attempt [NCS$^+$22] seeks to introduce sparsity in DP-trained large embedding models through post-processing `DP-SGD` gradients. However, this method falls short of providing the necessary DP guarantees. The closest effort to ours is the `DP-SGD` with exponential selection algorithm [ZMH21], which dynamically selects embedding buckets to update based on their gradient norms. However, their evaluation is limited to small-scale Word2Vec [MCCD13] models, with results mainly presented for a fixed large $\varepsilon = 30$. Furthermore, our experimental results in Section 4 demonstrate that their approach leads to a significant loss in utility when applied to large-scale recommendation tasks.

Additionally, prior research has also made efforts to improve the efficiency of DP-trained language models. For instance, [LTLH21] observes that embedding layers can significantly contribute to memory consumption during DP training of large language models. They therefore introduce ghost clipping to reduce the memory impact of embedding layers by avoiding per-example gradient generation. While our primary focus is on improving model efficiency, our methods also result in reduced memory usage as a secondary benefit due to gradient size reduction. Besides, [YNB$^+$22] proposes using parameter-efficient fine-tuning techniques in DP training but does not explore effective training of large embedding layers under DP: They freeze the embedding layers during DP fine-tuning and explore DP's compatibility with parameter-efficient fine-tuning methods, such as LoRA [HSW$^+$22] and Adapter [HGJ$^+$19], for attention layers. It is worth noting that, as discussed in Section 4.4, when LoRA is applied to embedding layers, the gradient size reduction achieved is inferior to the reductions introduced by our proposed methods.

In addition, it is worth mentioning that partition selection is also a technique for feature selection; its primary goal is to confidentially identify elements, along with their respective frequencies, that occur at least a specified number of times [KKMN09, DVGM22, GGK$^+$20, CGSS21, CWG22, SDH23]. However, in our current work, we introduce a new algorithm as it is more compatible with `DP-SGD`.

# B   Methodology

## B.1   DP Top-$k$ Selection

Assuming there are $c$ unique buckets in $L$, our goal is to return the top-$k$ frequent buckets from $L$ in a private way. In our scenario, each user's data can modify the count of a single bucket from a given feature by at most 1, ensuring that a user contributes to at most one bucket. Therefore, the $\ell_\infty$-sensitivity of the counting function for each feature is 1. To achieve DP top-$k$ selection, we

utilize the algorithm introduced in [DR21] (see Algorithm 2). The algorithm begins by counting the frequency $h \in \mathbb{N}^c$ of each bucket in the list. Next, it injects Gumbel noise into $h$ and subsequently returns the top-$k$ buckets based on the noisy frequency.

---

**Algorithm 2:** One-shot DP Top-$k$ Selection [DR21].

---

**Input** : The number of unique buckets $c$, a list of bucket occurrences of size $l$, $L \in [c]^l$, privacy budget $\varepsilon$, the number of selected buckets $k$.
**Output** : Top-$k$ frequent buckets selected, with DP, from $L$.

**1** Compute the bucket frequency in $L$;
**2 for** $i = 1$ **to** $l$ **do**
**3** $\quad \lfloor \; h[L[i]] \leftarrow h[L[i]] + 1$;
**4** Add Gumbel noise to the bucket frequency, $\hat{h} \leftarrow h + X$, where $X_i \sim \text{Gumbel}(1/\varepsilon)$;
**5** $r \leftarrow \arg\max_k\{\hat{h}_i : i \in [c]\}$, where $\arg\max_k$ returns the indices of the largest $k$ elements in a vector;
**6 Return** $r$

---

In our experiments, when selecting the top-$k$ buckets from all $p$ features, we distribute the privacy budget $\varepsilon$ and selection budget $k$ equally among the $p$ features. We then perform DP Top-$\text{int}(k/p)$ selection, allocating a budget of $\varepsilon/p$ for each feature. Finally, we concatenate the outputs from different features to obtain the final set of selected buckets. We set $\varepsilon = 0.01$ in our experiments and deduct the value from the total privacy budget accordingly as the budget for DP training.

## B.2 Memory Efficient Filtering

Implementing the computation of the contribution map and thresholding (Lines 6 and 8 in Algorithm 1) can naively take $O(c)$ time and space, since $v_i$'s are $c$-dimensional. This can be prohibitively expensive when $c$ is much larger than the size of the gradients. Here we sketch a different approach for computing the contribution map in time and space cost that is linear in the size of the gradient.

Let $\hat{V}_t = \sum_{i \in \mathcal{B}_t} [v_i]_{C_1}$. For all $j \in [c]$, we have that $\Pr[V_t[j] \geq \tau] = \Psi\left(\frac{\tau - \hat{V}_t[j]}{\sigma_1^2 \cdot C_1^2}\right)$, where $\Psi(t) := \Pr_{Z \sim \mathcal{N}(0,1)}[Z \geq t]$ is the survival function of the Gaussian distribution.

With this notation, a more efficient algorithm is as follows: For each $j$ such that $\hat{V}_t[j] \neq 0$, we sample the bits $V_t[j]$ accordingly. Let $c'$ be the number of coordinates $j$ such that $\hat{V}_t[j] = 0$ and for simplicity, we assume w.l.o.g. that these coordinates are contiguous. For sampling the other coordinates of $V_t$, we rely on the following generic approach for sampling a long binary vector, with each bit being $1$ with probability $p = \Psi(\tau/\sigma_1^2 C_1^2)$. Namely, we observe that the difference between the positions of two consecutive 1s is distributed as a geometric random variable $\text{Geom}(p)$, with probability mass function $p(1-p)^{k-1}$. Thus, we sample these indices directly by sampling the differences iteratively from $\text{Geom}(p)$, until the sum of these exceeds $c'$. This step runs in time that is linear in the number of false positives, which in expectation is $c'p$, which is proportional to the number of non-zeros in $G_t$.

## C Privacy Analysis

### C.1 Dominating Pairs for Privacy Analysis

The $e^\varepsilon$-*hockey stick divergence* between $P$ and $Q$ is given as $\mathscr{D}_{e^\varepsilon}(P||Q) = \sup_S P(S) - e^\varepsilon Q(S)$. Observe that a mechanism $M$ satisfies $(\varepsilon, \delta)$-DP if for all adjacent datasets $D \sim D'$ it holds that $\mathscr{D}_{e^\varepsilon}(M(D)||M(D')) \leq \delta$.

**Definition C.1** (Definition 7 in [ZDW22]). A pair $(P, Q)$ of random variables over $\Omega$ *dominates* a pair $(P', Q')$ of random variables over $\Omega'$ (denoted $(P', Q') \preceq (P, Q)$) if for all $\varepsilon \in \mathbb{R}$, it holds that $\mathscr{D}_{e^\varepsilon}(P'||Q') \leq \mathscr{D}_{e^\varepsilon}(P||Q)$.

*Remark.* The notion of "dominates" is symmetric, namely if $(P', Q') \preceq (P, Q)$ then $(Q', P') \preceq (Q, P)$. This follows because $\mathscr{D}_{e^\varepsilon}(Q||P) = (1 - e^\varepsilon) + e^\varepsilon D_{e^{-\varepsilon}}(P||Q)$.

To provide an example, we recall the Gaussian mechanism $M_{\text{Gauss}}^\sigma$, which takes as input vectors $x_1, \ldots, x_n \in \mathbb{R}^d$ with $\|x_i\|_2 \leq 1$ and returns $\sum_i x_i + \mathcal{N}(\mathbf{0}, \sigma^2 I_d)$, where $\sigma$ denotes the *noise multiplier*.

**Lemma C.2** (Gaussian mechanism). *For any adjacent datasets $D \sim D'$, it holds that $(M(D), M(D')) \preceq (\mathcal{N}(1, \sigma^2), \mathcal{N}(0, \sigma^2))$.*

We use two key behaviors of dominating pairs, namely under composition and Poisson subsampling.

**Composition.** For any mechanism $M_1$, and mechanism $M_2$, let $M = (M_1, M_2)$ denote the composed mechanism, namely, one that outputs $(y_1, y_2)$ sampled as $y_1 \sim M_1(D)$ and $y_2 \sim M_2(D; y_1)$.

**Lemma C.3** (Theorem 10 in [ZDW22]). *For any adjacent datasets $D$ and $D'$, if $(M_1(D), M_1(D')) \preceq (P_1, Q_1)$ and $(M_2(D; y), M_2(D'; y)) \preceq (P_2, Q_2)$ for all values of $y$, then $(M(D), M(D')) \preceq (P_1 \times P_2, Q_1 \times Q_2)$, where $P_1 \times P_2$ refers to the product distribution with marginals $P_1$ and $P_2$.*

**Poisson Subsampling.** For any $\gamma \in (0, 1]$, let $S_{\mathrm{Poi}}^{\gamma}$ denote the mechanism that takes as input a dataset of arbitrary size, and returns a dataset by including each datapoint with probability $\gamma$ i.i.d. at random.

**Lemma C.4** (Theorem 11 in [ZDW22]). *Let $M$ be a mechanism and $(P, Q)$ be a pair of distributions, such that for any adjacent datasets $D \sim D'$ with $D$ containing one more datapoint than $D'$, it holds that $(M(D), M(D')) \preceq (P, Q)$. Then for $\hat{M} = M \circ S_{\mathrm{Poi}}^{\gamma}$, it holds that $(\hat{M}(D), \hat{M}(D')) \preceq ((1 - \gamma)Q + \gamma P, Q)$.*

Finally, we recall the basic composition theorem of DP.

**Lemma C.5.** *If $M$ satisfies $(\varepsilon, \delta)$-DP and $M'$ satisfies $(\varepsilon', \delta')$-DP, then the composed mechanism $(M, M')$ satisfies $(\varepsilon + \varepsilon', \delta + \delta')$-DP.*

### C.2 Privacy Analysis of DP-SGD

We recall the privacy analysis of DP-SGD using privacy random variables, following [ACG+16], under Poisson sub-sampling of mini-batches. Observe that the computation of noisy gradient in each step of DP-SGD corresponds to an invocation of the Gaussian mechanism with noise multiplier of $\sigma$. Hence, by combining Lemmas C.2 to C.4, we have that for any adjacent datasets $D$ and $D'$ with $D$ containing one more datapoint than $D'$, it holds for $M$ being DP-SGD that $(M(D), M(D')) \preceq (P^{\times T}, Q^{\times T})$ where $P = (1 - \gamma)\mathcal{N}(0, \sigma^2) + \gamma\mathcal{N}(1, \sigma^2)$, $Q = \mathcal{N}(0, \sigma^2)$), and $T$ denotes the number of steps.

Thus, it suffices to choose $\sigma$ such that $\max\{\mathscr{D}_{e^{\varepsilon}}(P^{\times T} \| Q^{\times T}), \mathscr{D}_{e^{\varepsilon}}(Q^{\times T} \| P^{\times T})\} \leq \delta$ for desired privacy parameters $(\varepsilon, \delta)$.

### C.3 Privacy Analysis of DP-FEST

DP-FEST is simply the composition of two mechanisms: the first one being Algorithm 2 (that satisfies $(\varepsilon_1, 0)$-DP and DP-SGD with noise multiplier $\sigma$. Using Lemma C.5, and the above analysis of DP-SGD, it suffices to choose $\sigma$ such that $\max\{D_{e^{\varepsilon - \varepsilon_1}}(P^{\times T} \| Q^{\times T}), D_{e^{\varepsilon - \varepsilon_1}}(Q^{\times T} \| P^{\times T})\} \leq \delta$ for desired final privacy parameters of $(\varepsilon, \delta)$.

### C.4 Privacy Analysis of DP-AdaFEST

DP-AdaFEST can be analyzed essentially identically to DP-SGD. We observe that each step of DP-AdaFEST involves computing the private contribution map, and then the noisy gradient subject to the private contribution map. The private contribution map is computed via the Gaussian mechanism with noise multiplier $\sigma_1$ and the noisy gradient is also computed via the Gaussian mechanism with noise multiplier of $\sigma_2$. From [DRS19] (Corollary 3.3), it is known that the composition of two Gaussian mechanisms with noise multipliers of $\sigma_1$ and $\sigma_2$ respectively is equivalent to a Gaussian mechanism with noise multiplier $\sigma = (\sigma_1^{-2} + \sigma_2^{-2})^{-2}$, i.e., for $M_0 = (M_{\mathrm{Gauss}}^{\sigma_1}, M_{\mathrm{Gauss}}^{\sigma_2})$, it holds for any adjacent datasets $D \sim D'$ that $(M_0(D), M_0(D')) \preceq (\mathcal{N}(1, \sigma^2), \mathcal{N}(0, \sigma^2))$. Hence, by combining with Lemmas C.3 and C.4, we have that for any adjacent datasets $D$ and $D'$ with $D$ containing one more datapoint than $D'$, it holds for $M$ being DP-AdaFEST that $(M(D), M(D')) \preceq (P^{\times T}, Q^{\times T})$ where $P = (1 - \gamma)\mathcal{N}(0, \sigma^2) + \gamma\mathcal{N}(1, \sigma^2)$, $Q = \mathcal{N}(0, \sigma^2)$), and $T$ denotes the number of steps.

Thus, it suffices to choose $\sigma$ such that $\max\{\mathscr{D}_{e^{\varepsilon}}(P^{\times T} \| Q^{\times T}), \mathscr{D}_{e^{\varepsilon}}(Q^{\times T} \| P^{\times T})\} \leq \delta$ for desired privacy parameters $\varepsilon, \delta$.

## C.5 Numerical Privacy Accounting using Privacy Loss Distributions

Thus, in order to choose the noise parameters for all the above algorithms, the core primitive needed is to compute the smallest $\sigma$ such that $\max\{\mathscr{D}_{e^\varepsilon}(P^{\times T}\|Q^{\times T}), \mathscr{D}_{e^\varepsilon}(Q^{\times T}\|P^{\times T})\} \leq \delta$ for desired privacy parameters $\varepsilon, \delta$, where $P = (1-\gamma)\mathcal{N}(0, \sigma^2) + \gamma\mathcal{N}(1, \sigma^2), Q = \mathcal{N}(0, \sigma^2))$.

Several works have proposed numerical algorithms based on privacy loss distributions [KJH20, GLW21, GKKM22, DGK$^+$22] for privacy accounting and have in particular studied the specific pair $(P^{\times T}, Q^{\times T})$ above. In particular, we use the open-source implementation available in Google's DP library [Goo20] for this privacy accounting.

# D  Experimental Details and Other Results

## D.1  Experimental Details

In this section, we provide experimental details for recommendation tasks and NLU tasks.

### D.1.1  Click-Through Rate (CTR) Prediction Tasks

**The pCTR model's architecture.** The pCTR model we use comprises a total of six layers. The first layer utilizes an embedding layer to map each categorical feature to a dense feature vector. Appendix D.1.1 gives the vocabulary size of each categorical feature. We determine the embedding dimension using a heuristic rule of $\text{int}(2V^{0.25})$, where $V$ represents the number of unique tokens in each categorical feature. The resulting dense features are combined with the log-transformed integer features to create the first layer representation. We then apply four fully connected layers to this representation, each with an output dimension of $598$ and a ReLU activation function. Finally, we use another fully connected layer to compute the scalar prediction as the final output.

| Feature name | Vocabulary size | Feature name | Vocabulary size |
|---|---|---|---|
| categorical-feature-14 | $1,472$ | categorical-feature-27 | $27$ |
| categorical-feature-15 | $577$ | categorical-feature-28 | $1,550$ |
| categorical-feature-16 | $82,741$ | categorical-feature-29 | $44,262$ |
| categorical-feature-17 | $18,940$ | categorical-feature-30 | $10$ |
| categorical-feature-18 | $305$ | categorical-feature-31 | $5,485$ |
| categorical-feature-19 | $23$ | categorical-feature-32 | $2,161$ |
| categorical-feature-20 | $1,172$ | categorical-feature-33 | $3$ |
| categorical-feature-21 | $633$ | categorical-feature-34 | $56,473$ |
| categorical-feature-22 | $3$ | categorical-feature-35 | $17$ |
| categorical-feature-23 | $9,090$ | categorical-feature-36 | $15$ |
| categorical-feature-24 | $5,918$ | categorical-feature-37 | $27,360$ |
| categorical-feature-25 | $64,300$ | categorical-feature-38 | $104$ |
| categorical-feature-26 | $3,207$ | categorical-feature-39 | $12,934$ |

Table 3: Vocabulary size of 26 categorical features in the Criteo dataset.

**Hyper-parameters.** For `DP-SGD`, we fine-tune the clipping norm and report the best accuracy achieved. When evaluating `DP-FEST`, we adjust the hyper-parameter $k$, which represents the number of preserved top buckets, with values ranging from $100$ to $300,000$. Regarding `DP-AdaFEST`, we tune the following hyper-parameters:

▷ The ratio of noise added to the contribution map to the one added to the sparse gradient, $\sigma_1/\sigma_2$, with options of 0.1 to 10.

▷ The thresholding value $\tau \in \{0.5, 1.0, 5.0, 10.0, 20.0, 50.0, 100.0\}$.

▷ The clipping norm for gradient contribution $C_1 \in \{1.0, 5.0, 10.0\}$.

### D.1.2  NLU Tasks

**Datasets.** Our evaluation of NLU tasks uses three downstream classification tasks from the GLUE benchmark [WSM$^+$19]:

▷ The SST-2 dataset [SPW$^+$13] is a sentiment analysis dataset. The task is to predict whether a movie review is positive or negative.

▷ The QNLI dataset [RZLL16] is a question-answering dataset. Each example consists of a question and a corresponding sentence, and the task is to predict whether the sentence contains the answer to the question or not.

▷ The QQP dataset [IYW$^+$17] is a collection of question pairs that are labeled as either duplicate or not duplicate. The task is to predict whether a given pair of questions are duplicates or not.

**Hyper-parameters.** For `DP-SGD`, we fine-tune the clipping norm and report the best accuracy achieved. When evaluating `DP-FEST`, we adjust the hyper-parameter $k$, which represents the number of preserved top buckets, with values ranging from $1,000$ to $50,000$. Regarding `DP-AdaFEST`, we tune the following hyper-parameters:

▷ The ratio of noise added to the contribution map to the one added to the sparse gradient, $\sigma_1/\sigma_2$, with options of $0.1$ to $10$.

▷ The thresholding value $\tau \in \{50, 100, 200, 500, 1,000\}$.

▷ The clipping norm for gradient contribution $C_1 \in \{50.0, 100.0, 500.0\}$.

### D.2 More Experiment Results

#### D.2.1 Wall-Clock Time Reduction of Sparsity-Preserving `DP-SGD` Compared with `DP-SGD`

We conduct simulation experiments utilizing `tf.keras.layers.Embedding`[6], the sparse embedding API provided by Tensorflow that supports sparse lookup and sparse gradient computation, to assess the efficiency improvement of sparsity-preserving DP training compared to vanilla `DP-SGD`. Our simulations create an embedding layer with a fixed embedding dimension of $64$ and batch size of $1,024$. We investigate the impact of changing the vocabulary size $|V|$ on the wall-clock time improvement. Table 4 reports the training time for $100$ steps for `DP-SGD` and `DP-AdaFEST`, along with the wall-clock time reduction factor. We observe significant wall-clock time improvement in sparse updates, especially for larger vocabulary sizes: For instance, when the vocabulary size is 10M, we are able to achieve $> 150\times$ wall-clock time improvement over vanilla `DP-SGD`; For evaluation in Section 4, the Criteo models have a vocabulary of 1.7M, where our approach translates to a wall-clock time improvement ranging from 20 to $40\times$.

| Vocabulary Size | DP-SGD | Ours | Wall-Clock Time Reduction Factor |
|---|---|---|---|
| $1 \times 10^5$ | 0.985 | 0.324 | 3.045 |
| $2 \times 10^5$ | 1.676 | 0.312 | 5.376 |
| $1 \times 10^6$ | 6.607 | 0.318 | 20.746 |
| $2 \times 10^6$ | 13.09 | 0.323 | 40.515 |
| $5 \times 10^6$ | 31.326 | 0.332 | 94.340 |
| $1 \times 10^7$ | 62.473 | 0.353 | 176.760 |

Table 4: Wall-clock time improvement by our methods over `DP-SGD` under different vocabulary sizes.

This efficiency improvement comes from two sources:

▷ The cost of a sparse update (adding a sparse gradient tensor to the dense weight tensor) is smaller than that of a dense update, and this gap increases with the tensor size. For example, in the Tensorflow optimizers, `scatter_add()` is used when the gradient tensor is sparse to take advantage of this.

▷ The computation overhead of generating a dense tensor of Gaussian noises at each step becomes significant in large embedding layers and the cost also scales with the embedding layer size.

#### D.2.2 Recommendation Tasks

**Sparsity-preserving `DP-SGD` offers customizable options for balancing efficiency and utility.** The scatter plot in Figure 8 showcases how both `DP-AdaFEST` and `DP-FEST` present a range of

---

[6]https://www.tensorflow.org/api_docs/python/tf/keras/layers/Embedding

possibilities for effectively balancing utility and efficiency by leveraging sparsity-controlling parameters. We further note that both approaches yield an improved trade-off between privacy and utility compared to the previously suggested `DP-SGD` with exponential selection [ZMH21].

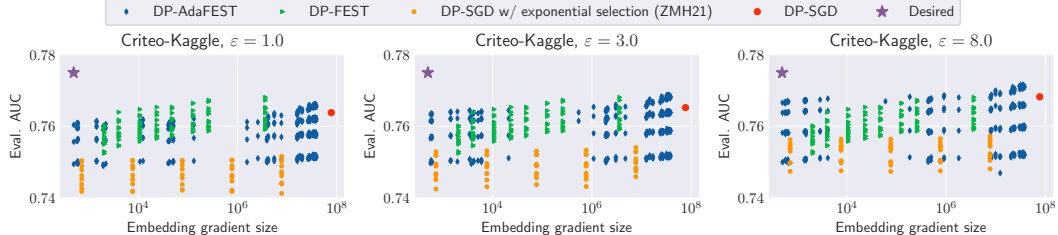

Figure 8: Comparison of the utility/efficiency trade-off between `DP-SGD`, `DP-SGD` w/ exponential selection [ZMH21], `DP-FEST`, and `DP-AdaFEST` on the Criteo-Kaggle dataset. `DP-FEST`, `DP-AdaFEST`, and `DP-SGD` w/ exponential selection can offer a more customized approach for meeting varying requirements of gradient sparsity and utility than `DP-SGD`, while `DP-AdaFEST` achieves the best trade-off.

**Effect of the streaming period on pCTR model's utility.** In Table 5, we present the evaluation AUC of Criteo-time-series under different streaming periods, with varying $\varepsilon$ values, for both vanilla `DP-SGD` and non-private training. Our results indicate that, for DP training, the AUC values increase as the streaming period increases, regardless of the value of $\varepsilon$ (i.e., $\varepsilon = 1.0$, $\varepsilon = 3.0$, and $\varepsilon = 8.0$). Conversely, non-private training results in similar AUC values across different streaming periods, which suggests that the model is less susceptible to distribution shifts in the data. This observation implies that DP training is more vulnerable to such shifts, thus explaining the effectiveness of `DP-AdaFEST`, which adapts to distribution shifts.

| Streaming period | $\varepsilon = 1.0$ | $\varepsilon = 3.0$ | $\varepsilon = 8.0$ | Non-private ($\varepsilon = \infty$) |
|---|---|---|---|---|
| 1 | 0.7313 | 0.7323 | 0.7345 | 0.7846 |
| 2 | 0.7313 | 0.7324 | 0.7346 | 0.7848 |
| 4 | 0.7314 | 0.7326 | 0.7352 | 0.7847 |
| 8 | 0.7316 | 0.7328 | 0.7352 | 0.7820 |
| 16 | 0.7318 | 0.7331 | 0.7356 | 0.7848 |
| 18 | 0.7320 | 0.7332 | 0.7358 | 0.7848 |

Table 5: Evaluation AUC of Criteo-time-series model under vanilla `DP-SGD` and non-private training with different streaming periods. The streaming period affects `DP-SGD` models but not non-private models.

**Joint impact of hyper-parameters in `DP-AdaFEST`.** In Figure 9, we present the interplay between the noise ratio $\sigma_1/\sigma_2$ and $\tau$ on the utility and embedding gradient size for the Criteo-Kaggle dataset. Our findings indicate that the optimal selection of hyper-parameters is critical for achieving a desirable trade-off between high utility and small gradient size. Specifically, we observe that darker colors on both heatmaps correspond to optimal hyper-parameter choices. Such choices are generally clustered in the middle-lower regions of the heatmaps, where we see a prevalence of high values of $\sigma_1/\sigma_2$ and medium values of $\tau$. These results suggest that the choice of hyper-parameters should be approached with great care and attention to detail, as it can have a significant impact on the performance of DP models.

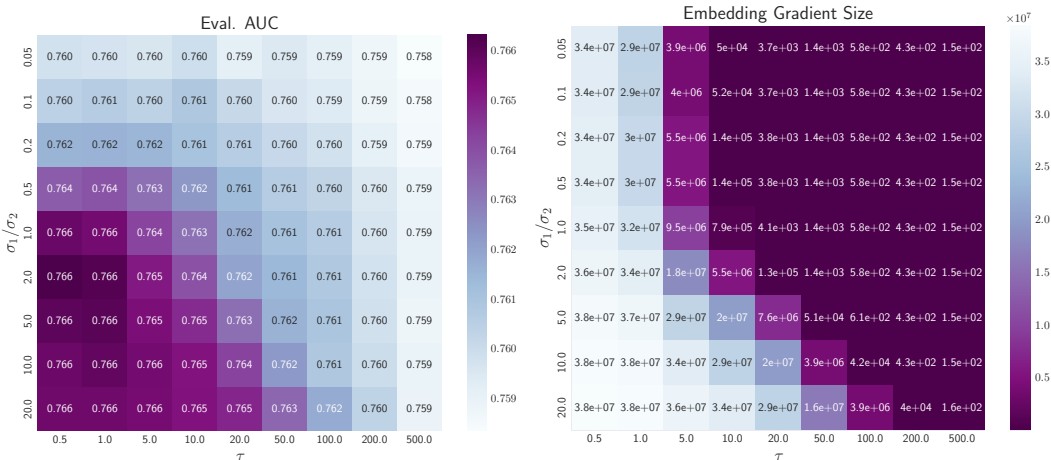

Figure 9: Heatmap illustrating the joint impact of the noise ratio $\sigma_1/\sigma_2$ and $\tau$ on the utility and embedding gradient size for the Criteo-Kaggle dataset ($\varepsilon = 1.0$).

### D.2.3 NLU Tasks

**Training word embedding layers in NLU improves accuracy.** As shown in Table 6, allowing gradient updates for word embedding layers in the NLU tasks improves the accuracy of DP training.

|  | SST-2 | QNLI |
|---|---|---|
| Non-private | 94.31% | 92.38% |
| DP-SGD, $\varepsilon = 1.0$ | 91.85% | 83.49% |
| DP-SGD, $\varepsilon = 1.0$ (embedding frozen) | 91.43% | 83.11% |
| DP-SGD, $\varepsilon = 3.0$ | 92.43% | 84.32% |
| DP-SGD, $\varepsilon = 3.0$ (embedding frozen) | 91.51% | 84.02 % |
| DP-SGD, $\varepsilon = 8.0$ | 92.87% | 86.34% |
| DP-SGD, $\varepsilon = 8.0$ (embedding frozen) | 92.66% | 86.28% |

Table 6: Accuracy comparison of non-private training and DP-SGD with and without frozen word embeddings on four language understanding datasets with varying levels of privacy budget ($\varepsilon$).

