# OpenReview forum: "Sparsity-Preserving Differentially Private Training of Large Embedding Models"
_NeurIPS.cc/2023/Conference — NeurIPS 2023 poster_

### Official Review · Reviewer_AuCL · 2023-07-06

**Soundness:** 3 good
**Presentation:** 2 fair
**Contribution:** 2 fair
**Rating:** 6
**Confidence:** 4

**Summary:**

This paper targets the learning of embedding models with DP-SGD.
The main idea is to utilize the sparsity of gradients to the embeddings.
If a mask indicating the sparsity of gradients is given, one can simply adapt DP-SGD by adding noises  only to the masked dimensions.
They propose two variants, DP-FEST and DP-AdaFEST, where in DP-FEST the mask is computed beforehand and in DP-AdaFEST the mask will be estimated with differential privacy by thresholding noisy version of the original gradients.
They evaluate the proposed methods on recommendation and language understanding tasks.

**Strengths:**

The problem investigated, i.e. learning embedding models through DP-SGD, is meaningful and the proposed method is very intuitive.

**Weaknesses:**

1. There seems to be a mismatch of scopes between the title and the rest of the paper. The sparsity of gradients with DP-SGD is only possible with the proposed scheme when there is naturally sparsity introduced by design, such as when each sample involves only a few rows in the entire embedding matrix (i.e. in the vocabulary) and the vocabulary contains many, many embeddings. It is important to have a title accurately reflecting the scope of the work. I would suggest, for example, Sparsity-Preserving Differentially Private Training of Embedding Vocabularies.



**Questions:**

See Weakness for some other comments.
1. It is a good practice to include the performances of non-private baselines for empirical experiments. While in many cases there can still be sizable gaps between DP methods and non-private baselines, it can help readers to understand intuitively how large the gaps remain to be.

2. While quite a few work improving DP-SGD are already mentioned in introduction, there are indeed some improvements of DP-SGD that are overlooked ([1-3]). While it is fine to focus on vanilla DP-SGD for experiments since it is still popular, I suggest to discuss these work as part of the background.

**reference:**

[1] Wang, Wenxiao, Tianhao Wang, Lun Wang, Nanqing Luo, Pan Zhou, Dawn Song, and Ruoxi Jia. "DPlis: Boosting Utility of Differentially Private Deep Learning via Randomized Smoothing." Proceedings on Privacy Enhancing Technologies 4 (2021): 163-183.

[2] Shamsabadi, Ali Shahin, and Nicolas Papernot. "Losing less: A loss for differentially private deep learning." (2021).

[3] Park, Jinseong, Hoki Kim, Yujin Choi, and Jaewook Lee. "Differentially Private Sharpness-Aware Training." arXiv preprint arXiv:2306.05651 (2023).

**Limitations:**

The authors addressed the limitations fairly.

---

> ### Author Rebuttal · Authors · 2023-08-09
>
> We genuinely appreciate the valuable feedback provided by the reviewer and have addressed them in a point-by-point manner below. We are more than willing to engage in further discussions with the reviewers should any follow-up questions arise.
>
> ### **Q1. A better title**
> > There seems to be a mismatch of scopes between the title and the rest of the paper. The sparsity of gradients with DP-SGD is only possible with the proposed scheme when there is naturally sparsity introduced by design, such as when each sample involves only a few rows in the entire embedding matrix (i.e. in the vocabulary) and the vocabulary contains many, many embeddings. It is important to have a title accurately reflecting the scope of the work. I would suggest, for example, Sparsity-Preserving Differentially Private Training of Embedding Vocabularies.
>
> **A**: We appreciate the reviewer’s comment.  In the revision, we will change the title to "Sparsity-Preserving Differentially Private Training of Large Embedding Models", to more accurately reflect the scope of our work.
>
> ### **Q2. Performance of the non-private baseline**
> > It is a good practice to include the performances of non-private baselines for empirical experiments. While in many cases there can still be sizable gaps between DP methods and non-private baselines, it can help readers to understand intuitively how large the gaps remain to be.
>
> **A**:  Tables 2 & 3 in the Appendix of our submission contain the non-private baseline results for Criteo and NLU tasks.  If the reviewer thinks it’d be helpful, we could consider adding in the revision the following table which contains the non-private baseline numbers for all datasets.
>
> | Task                                     | Non-private baseline |
> |------------------------------------------|----------------------|
> |     **Recommendation task** (metric: AUC)    |                      |
> | Criteo-Kaggle                            | 0.8063               |
> | Criteo-time-series (streaming period=1)  | 0.7846               |
> | Criteo-time-series (streaming period=2)  | 0.7848               |
> | Criteo-time-series (streaming period=4)  | 0.7847               |
> | Criteo-time-series (streaming period=8)  | 0.7820               |
> | Criteo-time-series (streaming period=16) | 0.7848               |
> | Criteo-time-series (streaming period=18) | 0.7848               |
> |  **NLU task** (metric: accuracy)  |                      |
> | SST-2                                    | 94.31%               |
> | QNLI                                     | 92.38%               |
> | QQP                                      | 91.67%               |
>
> ### **Q3. New references for improvements of vanilla DP-SGD**
>
> **A**: We thank the reviewer for sharing the new references that improve vanilla DP-SGD; we will add them in the revision.

---

> > ### Comment · Reviewer_AuCL · 2023-08-18
> >
> > Thanks for the rebuttal. Since I have no major concern regarding this submission, I think my rating still reflects my assessment of this work. I will keep the score as it is for now.
> >
> > Have a good day!

---

> > > ### Author Response · Authors · 2023-08-18
> > >
> > > We appreciate the reviewer's response and it's great to hear that the reviewer has no major concern about our work. We are also more than willing to continue the discussion if the reviewer has any minor points they'd like to bring up.
> > >
> > > Happy a good day too!

---

### Official Review · Reviewer_KMWx · 2023-07-10

**Soundness:** 2 fair
**Presentation:** 1 poor
**Contribution:** 1 poor
**Rating:** 3
**Confidence:** 4

**Summary:**

This work aims at improving the performance of DP-SGD on models with a large embedding layer. The main idea is zeroing out the insignificant coordinates such that the amount of added Gaussian noise is reduced. The proposed methods achieve substantial gradient size reduction with marginal performance loss compared with the vanilla DP-SGD.

**Strengths:**

1. Differential privacy is an important topic in the study of privacy and security. Making DP more practically useful is a main challenge in the current research.
2. This work clearly clarify the background and its methodology, the paper is easy to follow.

**Weaknesses:**

1. The proposed method is similar to the sparse technologies in DP. There is a long line of works studying sparse technology and DP selection, many of them have also studied large embedding layer in the language models. This work is not properly placed in the contemporary literature.
2. Applying LoRA to the embedding layer can also reduce the gradient size and obtain similar advantages, it would be interesting to compare the proposed methods to such low-rank methods.

**Questions:**

1. What does "best" mean in the term "best gradient size reduction"?
2. According to my knowledge, if the sparsity pattern is changing consistently, it is hard to take advantage of this property and accelerate the computation. Please correct me if this is wrong for specific hardware, e.g. TPU.
3. Please state the accuracy of the baseline in the main text. This is crucial for the evaluation of the conducted experiments.

---

> ### Author Rebuttal · Authors · 2023-08-09
>
> We genuinely appreciate the valuable feedback provided by the reviewer and have addressed them in a point-by-point manner below. We are more than willing to engage in further discussions with the reviewers should any follow-up questions arise.
>
> ### **Q1. Missing placement of contribution?**
> > The proposed method is similar to the sparse technologies in DP. There is a long line of works studying sparse technology and DP selection, many of them have also studied large embedding layer in the language models. This work is not properly placed in the contemporary literature.
>
> **A**: Yes, there has been extensive work on DP selection. In contrast, to the best of our knowledge, there is no prior work that studies the *sparse DP training of large embedding layers in recommender systems and language models*. If the reviewer has pointers to any such references, please let us know and we are happy to incorporate them.
>
> ### **Q2. Applicability of LoRA to embedding layers**
> > Applying LoRA to the embedding layer can also reduce the gradient size and obtain similar advantages, it would be interesting to compare the proposed methods to such low-rank methods.
>
> **A**: We appreciate the reviewer’s suggestion. However, applying LoRA to the embedding layers is not a common practice due to the following reasons:
> 1. LoRA was introduced as a method to efficiently adapt matrices of dimensions $n \times d$ in language models by utilizing a rank-$r$ approximation (the initial use case considers the attention layers where $n=d$). The rank-$r$ approximation helps in reducing the memory requirements by a factor of $m \times d/(m+d)*r < min(m,d)/r$. However, in the case of embedding layers, where $n$ represents the vocabulary size and $d$ denotes the embedding dimensionality, a notable disparity exists: the vocabulary size $n$ is typically very large (>1M in the evaluated recommendation task and ~50,000 in the evaluated NLU task), while the embedding dimensionality $d$ is in the hundreds. Consequently, **the potential for improvements using LoRA in this context is limited**.
> 2. For private training of the embedding layer, using DP-AdaFEST we could still benefit from the efficient embedding lookup (i.e., row fetching operations) via customized APIs. However, LoRA would **not be able to leverage these APIs** as it requires relatively expensive matrix multiplication.
> 3. While **LoRA needs to adapt a pre-trained model**, our algorithm works in both pre-training and fine-tuning, as demonstrated in the recommender system and language model experiments, respectively.
>
> To more effectively demonstrate the arguments above, the table below compares the best embedding gradient size reductions achieved by AdaFEST and LoRA against DP-SGD for SST-2 with $\epsilon=1.0$. We vary LoRA's rank $r$ from {4, 8, 16, 32, 64, 128}. AdaFEST consistently outperforms LoRA in gradient size reduction at similar utility levels. Additionally, AdaFEST allows effective utilization of reduced gradients through customized APIs, while the implications of LoRA's reduction remain unclear.
>
> | **Utility loss compared to DP-SGD** | **Best gradient size reduction, AdaFEST** | **Best gradient size reduction, LoRA** |
> |---|---|---|
> | 0.001 | 17.41x | 5.91x |
> | 0.005 | 62.14x | 23.64x |
> | 0.01 | 62.14x | 47.28x |
>
> ### **Q3. Speedup for changing sparsity patterns?**
> > According to my knowledge, if the sparsity pattern is changing consistently, it is hard to take advantage of this property and accelerate the computation. Please correct me if this is wrong for specific hardware, e.g. TPU.
>
> **A**: Leveraging sparsity efficiently often requires specialized algorithms and hardware capable of handling sparse data. Fortunately, while the sparsity exploited by our algorithm is dynamic, it follows a consistent structure that only a small number of **rows** of the gradient of the embedding layer are non-zero. This (dynamic) structure can be efficiently exploited by modern accelerators that provide dedicated implementation for embedding lookups.
>
> For example, Google Cloud TPUs can efficiently exploit such dynamic sparsity patterns. Appendix C.2 demonstrates significant wallclock time improvement using TPUEmbedding (please refer to footnote 6 in our Appendix C.2.1 for the link), the sparse embedding API provided by Google Cloud TPUs. This showcases TPUs' proficiency in handling sparse operations with compressed representations and efficient memory access patterns, proving their ability to make use of the dynamic sparsity patterns here.
>
> ### **Q4. What does "best" mean in the term "best gradient size reduction"?**
>
> **A**:  Our experiments explore various combinations of hyperparameters (Section 4.4 has the details) and we reported the maximum gradient size reduction achieved at different utility requirements. We will clarify this in the revision.
>
> ### **Q5. Please state the accuracy of the baseline in the main text**
>
> **A**:  We appreciate the great suggestion. We will report the accuracy in the main text in the revision.

---

> > ### Comment · Reviewer_KMWx · 2023-08-15
> > **Response by Reviewer**
> >
> > Thank you to the authors for providing detailed responses.
> >
> > My concerns regarding Q2 and Q4 have been addressed. I'd like to further clarify Q1, which is my major concern, and I'm not fully convinced by the response to Q3. Additionally, since the authors haven't updated the baselines' accuracies, I'm unable to evaluate some experimental results as stated in Q5.
> >
> > ## Q1:
> >
> > The authors have positioned this paper as the pioneering work in studying DP networks with large embedding layers.
> >
> > *Line 48 - 49*
> > > ...this study is the first to address the technical challenges of applying DP-SGD to large embedding models.
> >
> > While DP on large language models has been widely researched, with many previous works cited in this paper (e.g., [YNB+ 22] and [LTLH21]), it's worth noting that LTLH21 argued that DP full fine-tuning might outperform non-private fine-tuning. Thus, the above claim appears misleading or exaggerated. Moreover, this work's experimental settings, such as utilizing the RoBERTa network and certain language datasets, have been previously explored. Hence, this paper is not treading new ground.
> >
> > I perceive the primary focus of this work as preserving sparsity. From an abstract standpoint, the gradient of the embedding layer could be viewed as a vector with mostly zero elements. Numerous studies have tackled sparse vectors in the context of DP. I see the authors also agree with that, so I won't take the time to look up and copy the links. However, I find a gap in the introduction of relevant methods, and it is unclear why this specific form of DP-AdaFEST is chosen.
> >
> > The concept of sparing the addition of noise to large embedding layers dates back at least two years [1].
> >
> > In summary, preserving the sparsity of the embedding layer has been proposed before. The technology for preserving sparsity, e.g. DP selection, has been extensively studied. There are many previous works studying applying DP to large embedding models. As a result, I believe this work lacks novelty and a more thorough introduction of related works is needed.
> >
> > ## Q3
> >
> > Upon examining the wall-clock time reduction experiments (Appendix C.2.1), the results were astonishing: the proposed method appears to be approximately 200 times faster. My understanding of computation acceleration references cited here suggests that sparse gradients can be computed faster than dense gradients, thanks to sparse matrix multiplication. However, in the context of this paper, sparse gradients are computed initially, with the only distinction being whether sparsity is preserved during nosification. This preservation would only affect later parameter updates, amounting to mere matrix addition. I fail to see why this can lead to such a leap in computational speed.
> >
> > I wish to emphasize that sparsity is often exploited for computation acceleration due to sparse matrix multiplication. However, the proposed method doesn't involve this operation. Could the author elaborate on how preserving sparsity during nosification substantially boosts computational efficiency?
> >
> > ## Q5
> >
> > Please update the baseline accuracy during the discussion period. I believe the numbers should have been saved, and therefore no need to rerun all the experiments. Without this information, it is impossible to evaluate whether the proposed method is comparing with a reasonable baseline.
> >
> > [1] Wide Network Learning with Differential Privacy. Huangyu Zhang et al., 2021

---

> > > ### Author Response · Authors · 2023-08-18
> > > **Response to follow-up questions (1/3)**
> > >
> > > We appreciate the reviewer for taking the time to read our response and sharing follow-up concerns.
> > >
> > > ### **Q1. Missing placement of contribution?**
> > >
> > > > While DP on large language models has been widely researched, with many previous works cited in this paper (e.g., [YNB+ 22] and [LTLH21]), it's worth noting that LTLH21 argued that DP full fine-tuning might outperform non-private fine-tuning. Thus, the above claim appears misleading or exaggerated. Moreover, this work's experimental settings, such as utilizing the RoBERTa network and certain language datasets, have been previously explored. Hence, this paper is not treading new ground.
> > >
> > > **A**: We appreciate the reviewer for the pointers to the related work and for acknowledging that we've already discussed them in our submission. We hope to further clarify that the focus of our work is significantly different from [YNB+ 22] and  [LTLH21].
> > >
> > > [YNB+ 22] didn’t study how to effectively train large embedding layers under DP; they froze the embedding layers in DP-fine tuning and investigated DP’s compatibility with parameter-efficient fine-tuning methods for attention layers such as LoRA and Adapter. As we noted in the response to your Q2, when applied to embedding layers, the gradient size reduction introduced by LoRA is inferior to our proposals.
> > >
> > > [LTLH21] observed that embedding layers can significantly contribute to memory consumption during DP training of large language models. To address this, they applied ghost clipping (see Sec 4.2 in their paper) to avoid generating per-example gradients, thereby reducing the memory impact of embedding layers during training. We'd like to highlight the distinctions between our work and [LTLH21]:
> > > - **Primary goal**: Our primary goal is model efficiency, whereas [LTLH21] primarily targets memory reduction. However, please note that our methods also lead to reduced memory usage as a secondary benefit, because of the reduction in gradient size.
> > > - **Specific strategies**: We deliberately induce sparsity within embedding layers, whereas [LTLH21] focuses on optimizing per-example gradient operations to save memory.
> > > - **Beyond NLP models**: We focus on the sparsity of large embedding layers. While this benefits many transformer-based NLP models, an equally significant contribution of our study is in the systematic evaluations on recommender systems, where the embedding layers could occupy up to 90% of the model weights, and techniques like LoRA cannot be easily applied because there are no public pre-trained models.
> > >
> > > > I perceive the primary focus of this work as preserving sparsity. From an abstract standpoint, the gradient of the embedding layer could be viewed as a vector with mostly zero elements. Numerous studies have tackled sparse vectors in the context of DP. I see the authors also agree with that, so I won't take the time to look up and copy the links. However, I find a gap in the introduction of relevant methods, and it is unclear why this specific form of DP-AdaFEST is chosen.
> > >
> > > Yes, we indeed agree with the reviewer that there are numerous studies of the abstract problems of sparse vectors in DP. Our main contribution is applying such techniques to **large embedding models**, and **systematic evaluations** that demonstrate practical benefits in large-scale models in recommender systems and language models.
> > >
> > > We will expand our discussion of related work to fill the gap. This specific form of DP-AdaFEST was chosen to balance simplicity (minimum modification to existing DP-SGD privacy accounting) and practical performance. But we leave the extensive comparison of different sparse-preserving DP mechanisms to future work.
> > >
> > > > The concept of sparing the addition of noise to large embedding layers dates back at least two years [1].
> > >
> > > We appreciate this pointer, and will  try to report comparison results before the end of the rebuttal period for completeness. However, we respectfully hold a different opinion regarding the reviewer’s assessment of [1] tackling *“large”* embedding models, as [1] only evaluated with a small Word2Vec model:  they use an embedding layer with vocabulary size $|V|$=1000 and embedding dimension $d$=100, resulting in an embedding layer of size 100,000 (only twice the size of the simple LeNet-5 architecture for digit classification). While in our evaluation, the recommendation task involves the usage of $|V|$=1.7M and total embedding params of 77M, and the NLU task has $|V|$=50265 and $d$=768, and total embedding params of 38M. In summary, the models trained in our work are multiple orders of magnitude larger than [1], and significantly more representative of *large* embedding models used in practice.

---

> > > > ### Author Response · Authors · 2023-08-18
> > > > **Response to follow-up questions (2/3)**
> > > >
> > > > ### **Q3. Speedup from customized API**
> > > >
> > > > > Upon examining the wall-clock time reduction experiments (Appendix C.2.1), the results were astonishing: the proposed method appears to be approximately 200 times faster ... in the context of this paper, sparse gradients are computed initially, with the only distinction being whether sparsity is preserved during nosification. This preservation would only affect later parameter updates, amounting to mere matrix addition. I fail to see why this can lead to such a leap in computational speed.
> > > >
> > > > **A**: We appreciate the reviewer’s comment. First, we would like to clarify that the comparison presented in Appendix C.2.1 were comparing `dense-backpropagation + dense-update` against `sparse-backpropagation + sparse-update`. We acknowledge the reviewer's observation that vanilla DP-SGD indeed entails `sparse-backpropagation + dense-update`. **We apologize for the oversight and have rectified the comparison** accordingly as follows:
> > > >
> > > > Specifically, we fix the embedding dimension to 64 and batch size to 1024, and vary the vocabulary size. We report the training time for 100 steps for `sparse-backpropagation + dense-update` and `sparse-backpropagation + sparse-update`, along with the wall-clock time reduction factor. As shown, we are still able to observe significant wall-clock time improvement in sparse updates, especially for larger vocabulary sizes.
> > > >
> > > > | Vocabulary size | `sparse-backpropagation + dense-update` (Vanilla DP-SGD) | `sparse-backpropagation + sparse-update` (Ours) | Wall-clock time reduction factor |
> > > > |:---:|:---:|:---:|:---:|
> > > > | 100,000 | 0.985 | 0.324 | 3.045 |
> > > > | 200,000 | 1.676 | 0.312 | 5.376 |
> > > > | 1,000,000 | 6.607 | 0.318 | 20.746 |
> > > > | 2,000,000 | 13.09 | 0.323 | 40.515 |
> > > > | 5,000,000 | 31.326 | 0.332 | 94.340 |
> > > > | 10,000,000 | 62.473 | 0.353 | 176.76 |
> > > >
> > > > > I wish to emphasize that sparsity is often exploited for computation acceleration due to sparse matrix multiplication. However, the proposed method doesn't involve this operation. Could the author elaborate on how preserving sparsity during nosification substantially boosts computational efficiency?
> > > >
> > > > Thanks for the insightful question. The efficiency improvement of the updated comparison between `sparse-backpropagation + sparse-update` and `sparse-backpropagation + dense-update` mainly comes from two sources:
> > > > - The cost of sparse update (adding a sparse gradient tensor to the dense weight tensor) is still smaller than dense update, and this gap increases with the tensor size. For example, in the Tensorflow optimizers, `scatter_add()` is used when the gradient tensor is sparse to take advantage of this.
> > > > - The computation overhead of generating a dense tensor of Gaussian noises at each step becomes significant in large embedding layers and the cost also scales with the embedding layer size.
> > > >
> > > > ### **Q5. Provide baseline accuracy**
> > > > > Please update the baseline accuracy during the discussion period. I believe the numbers should have been saved, and therefore no need to rerun all the experiments. Without this information, it is impossible to evaluate whether the proposed method is comparing with a reasonable baseline.
> > > >
> > > > **A**: We appreciate the comment and provide the results below. Please refer to Sec 4.1.1 in our submission for model details.
> > > > Note that for the NLU task, our implementation was based on the DP-LoRA fine-tuning codebase by [YNB+ 22]. The difference is that we enable the training of embedding layers during model fine-tuning (as indicated around line 233 in our submission). The backbone model is RoBERTa-base.
> > > >
> > > > | Task | Non-private baseline | eps=1.0 | eps=3.0 | eps=8.0 |
> > > > |---|---|---|---|---|
> > > > |  | **Recommendation task (metric: AUC)** |  |  |  |
> > > > | Criteo-Kaggle | 0.8063 | 0.7670 | 0.7706 | 0.7758 |
> > > > |  | **NLU task (metric: accuracy)** |  |  |  |
> > > > | SST-2 | 94.31% | 91.85% | 92.43% | 92.87% |
> > > > | QNLI | 92.38% | 86.02% | 86.73% | 87.91% |
> > > > | QQP | 91.67% | 83.49% | 84.32% | 86.34% |
> > > >
> > > > Once again, we value the feedback from the reviewer and remain available for further discussion should any new questions arise.

---

> > > > > ### Author Response · Authors · 2023-08-20
> > > > > **Response to follow-up questions (3/3)**
> > > > >
> > > > > > The concept of sparing the addition of noise to large embedding layers dates back at least two years [1].
> > > > >
> > > > > We further provide the results for comparing our DP-AdaFEST with [1] on the Criteo-Kaggle benchmark. We chose this benchmark for demonstration because it has more embedding parameters (77M) than NLU tasks (38M). As shown, DP-AdaFEST consistently achieves higher gradient size reduction than [1]; also unlike [1], which struggles to compress gradients when the utility loss is minimal (say ≤ 0.005), DP-AdaFEST manages to achieve substantial reduction even under these circumstances.
> > > > >
> > > > > | Utility loss compared to DP-SGD | Best gradient size reduction, eps=1.0 (ours) | Best gradient size reduction, eps=1.0 ([1]) | Best gradient size reduction, eps=3.0 (ours) | Best gradient size reduction, eps=3.0 ([1]) | Best gradient size reduction, eps=8.0 (ours) | Best gradient size reduction, eps=8.0 ([1]) |
> > > > > |---|---|---|---|---|---|---|
> > > > > | 0.002 | **1.52** | 1 | **2.15** | 1 | **1.73** | 1 |
> > > > > | 0.005 | **484132.67** | 1 | **509176.32** | 1 | **476695.07** | 1 |
> > > > > | 0.01 | **484132.67** | 43122.86 | **509176.32** | 24430.82 | **486228.97** | 45.07 |
> > > > > | 0.02 | **1246036.81** | 183580.21 | **1246947.45** | 183580.21 | **1247423.67** | 183580.21 |
> > > > >
> > > > > Furthermore, as we explained in [our previous reply](https://openreview.net/forum?id=sqTcCXkG4P&noteId=6YlVg1LbDe), the evaluation conducted by [1] only focused on a small Word2Vec model with 100,000 embedding parameters. In contrast, our evaluation uses a recommendation task with 77 million embedding parameters, as well as an NLU task with 38 million embedding parameters. Therefore, our empirical study is more representative of *large* embedding models used in practice.

---

> > > > > > ### Comment · Reviewer_KMWx · 2023-08-21
> > > > > > **Response by Reviewer**
> > > > > >
> > > > > > Thanks for providing additional results and further clarification.
> > > > > >
> > > > > > ## Q1
> > > > > >
> > > > > > The authors may not fully understand my concern in Q1. While empirical evidence showing that your proposed method works better is valuable, this alone does not substantiate the claim: “this study is the first to address the technical challenges of applying DP-SGD to large embedding models.” I’m cautious with such a claim as it may misguide follow-up works in narrowing their literature review, possibly hindering the overall community. I suggest that the authors specify their definition of the general technical challenges of applying DP-SGD. Once provided, I will discuss with other reviewers and the area chair to evaluate if this claim is reasonable.
> > > > > >
> > > > > > Additionally, LTLH21 proposed the ghost clip to reduce the memory footprint of generating per-example gradient. Their ultimate goal, like yours, includes increasing throughput. [1] is more of a theoretical study, and the use of a relatively smaller network doesn’t imply that their work isn’t applicable to larger embedding layers or that your work is the first in this area. They’ve already noted the inherent sparsity of the embedding layer’s gradient and attempted to address this problem. I still believe that many previous works have attempted to address the technical challenges of applying DP-SGD to large embedding networks, some also take the same perspective of optimization efficiency.
> > > > > >
> > > > > > ## Q3
> > > > > >
> > > > > > The result is still a bit surprising, but now I’m convinced that preserving sparsity can update the embedding layer more efficiently. My remaining question pertains to how this improvement is reflected in the overall model optimization. Since the embedding layer may be a small part of the large-scale network, and the dense update to the embedding layer is still applied when using the Adam optimizer, I’m not sure if solely preserving sparsity will obviously enhance the overall optimization efficiency of large-scale networks with large embedding layers. Can you provide more clarity on this? Also, the writing style of this work could be improved to present the final profit more transparently.
> > > > > >
> > > > > > ## Q5
> > > > > >
> > > > > > This concern has been addressed.

---

> > > > > > > ### Author Response · Authors · 2023-08-21
> > > > > > > **Follow-up: Our Contribution & Speedup from Customized API**
> > > > > > >
> > > > > > > We appreciate the response and we are glad that the reviewer’s Q5 is addressed. Below we provide a point-wise response to the reviewer’s remaining concerns.
> > > > > > >
> > > > > > > ### **Q1. Missing placement of contribution?**
> > > > > > > > I suggest that the authors specify their definition of the general technical challenges of applying DP-SGD. Once provided, I will discuss with other reviewers and the area chair to evaluate if this claim is reasonable.
> > > > > > >
> > > > > > > **A**: We appreciate the reviewer’s suggestion and specify the technical challenge of applying DP-SGD to large embedding models: we specifically target the issue of diminished gradient sparsity encountered when employing DP-SGD with large embedding models due to the addition of dense noise; *this issue represents a significant practical obstacle for leveraging, during DP training, the hardware accelerators commonly used in non-private training of large embedding models*. Our study demonstrates, through experiments on real-world large embedding models (comprising over 30 million embedding parameters), that significant reduction in gradient size can be accomplished during differentially private training with negligible loss in utility.
> > > > > > >
> > > > > > > We also appreciate the reviewer’s comments on the related work, and will incorporate them into our final revision.
> > > > > > >
> > > > > > > ### **Q3. Speedup from customized API**
> > > > > > > > My remaining question pertains to how this improvement is reflected in the overall model optimization. Since the embedding layer may be a small part of the large-scale network, and the dense update to the embedding layer is still applied when using the Adam optimizer, I’m not sure if solely preserving sparsity will obviously enhance the overall optimization efficiency of large-scale networks with large embedding layers.
> > > > > > >
> > > > > > > **A**: Thanks for the question. In our experiments on recommendation tasks, embedding layers (**77M**) are the majority of model weights (**78M**). Thus, enhancing efficiency through the sparsification of gradient updates in these layers should play a pivotal role in driving the overall performance enhancement.
> > > > > > >
> > > > > > > For NLU tasks, while it is true that the embedding layer (~30M) may be a small part of the whole network, it's important to note that our proposed methodology seamlessly integrates with other parameter-efficient fine-tuning techniques applicable to various layers within the model. For instance, consider the scenario where LoRA is implemented for attention layers (the implementation we used in our experiments), there will be a remarkable reduction in the trainable parameters within the attention layers, bringing the number down to **<1M** for the RoBERTa model (as indicated in Table 2 of the LoRA paper). Even in such cases, the benefits derived from sparsity within the embedding layers would continue to exert a dominant influence on efficiency enhancement.
> > > > > > >
> > > > > > > > and the dense update to the embedding layer is still applied when using the Adam optimizer…
> > > > > > >
> > > > > > > We appreciate this comment. However, it's important to note that users typically have control over their choice of optimizers, and thus they should be able to apply a sparse gradient-friendly optimizer specifically for the embedding layer (and applying different optimizers for different layers is implementation-wise simple for most deep learning frameworks).
> > > > > > >
> > > > > > > > Also, the writing style of this work could be improved to present the final profit more transparently.
> > > > > > >
> > > > > > > Thanks for the suggestion. We will incorporate your feedback and improve the writing in the revision.

---

### Official Review · Reviewer_ah8R · 2023-07-13

**Soundness:** 3 good
**Presentation:** 3 good
**Contribution:** 2 fair
**Rating:** 5
**Confidence:** 3

**Summary:**

This paper considers an interesting and less-studied aspect of DP-SGD: Applying naively to embedding models can destroy gradient sparsity, reducing training efficiency.
To address this issue, the paper proposes two algorithms DP-FEST and DP-AdaFEST that apply DP while maintaining gradient sparsity during the training of large embedding models.

**Strengths:**

The authors did a great job motivating the problem in the introduction. The question addressed in this work is of great importance and a reasonable answer to it can potentially have significant consequences, as evidenced by the experiments.

**Weaknesses:**

The paper lacks any rigorous analysis of the proposed algorithms. More precisely, there is no privacy analysis of DP-FEST and DP-AdaFEST.
Instead, the authors only mentioned (line 169-170):  "In particular, the privacy cost of a single iteration is equivalent to that of the composition of two Gaussian mechanisms with noise scale $\sigma_1$
and $\sigma_2$ respectively, which in turn is equivalent to the privacy cost of a single Gaussian mechanism of noise scale $\sigma = (\sigma_1^{-2}+\sigma_2^{-2})^{-1/2}$." Why not using this result to determine a formal privacy analysis of the algorithm?

**Questions:**

1. Is there any gap between the privacy guarantee of SP-SGD and that of DP-AdaFEST? If so, can the authors quantify it?


**Limitations:**

Yes!

---

> ### Author Rebuttal · Authors · 2023-08-09
>
> We genuinely appreciate the valuable feedback provided by the reviewer and have addressed them below. We are more than willing to engage in further discussions with the reviewers should any follow-up questions arise.
>
> ### **Q1. Lack of rigorous privacy analysis**
> > The paper lacks any rigorous analysis of the proposed algorithms. More precisely, there is no privacy analysis of DP-FEST and DP-AdaFEST. Instead, the authors only mentioned (line 169-170): "In particular, the privacy cost of a single iteration is equivalent to that of the composition of two Gaussian mechanisms with noise scale $\sigma_1$ and $\sigma_2$ respectively, which in turn is equivalent to the privacy cost of a single Gaussian mechanism of noise scale $\sigma = (\sigma_1^{-2} + \sigma_2^{-2})^{-1/2}$." Why not using this result to determine a formal privacy analysis of the algorithm? Is there any gap between the privacy guarantee of SP-SGD and that of DP-AdaFEST? If so, can the authors quantify it?
>
> **A**:  Please note that a detailed privacy analysis was not included since we felt it is standard.  Indeed, the analysis is near-identical to that of DP-SGD and involves repeated application of the sub-sampled Gaussian mechanism with scale $\sigma$. In DP-AdaFEST, we have a repeated application of a sub-sampled mechanism where the inner mechanism involves application of two Gaussian mechanisms with noise scales $\sigma_1$ and $\sigma_2$. It is well-known (e.g., [1, Corollary 3.3]) that such an inner mechanism is privacy-wise equivalent to an application of a single Gaussian mechanism with noise scale $\sigma = (\sigma_1^{-2} + \sigma_2^{-2})^{-1/2}$, and thus, **DP-AdaFEST is privacy-wise equivalent to a sub-sampled Gaussian mechanism with scale $\sigma$**.
>
> The privacy analysis for DP-FEST is also straightforward, since it involves the composition of the first frequency filtering step and the second training step which is privacy-wise equivalent to a sub-sampled Gaussian mechanism.
>
> We thank the reviewer again for raising this point and we will add these details in the revision.
>
> **References**:
>
> [1] Jinshuo Dong, Aaron Roth, Weijie J. Su. Gaussian Differential Privacy.

---

> > ### Author Response · Authors · 2023-08-19
> > **Follow-up**
> >
> > Dear reviewer, did our explanation clarify your question on the privacy analysis? Since this seems to be the main concern from the reviewer, we would like to make sure we addressed it before the rebuttal deadline ends. To summarize the argument with more details:
> > * Both DP-SGD and DP-AdaFEST are performing repeated application of an “inner mechanism” on sub-sampled batches of data. The “inner mechanism” is a single Gaussian mechanism in case of DP-SGD, and is application of two Gaussian mechanisms in case of DP-AdaFEST.
> > * It is known from [1, Corollary 3.3] that the privacy loss random variable corresponding to the composition of two Gaussian mechanisms with noise scales $\sigma_1$ and $\sigma_2$ is identical to the privacy loss random variable of a single Gaussian mechanism with noise scale $\sigma = (\sigma_1^{-2} + \sigma_2^{-2})^{-1/2}$.
> > * Hence, the privacy guarantee of DP-AdaFEST with noise scales $\sigma_1$ and $\sigma_2$ is equivalent to the privacy guarantee of DP-SGD with noise scale $\sigma = (\sigma_1^{-2} + \sigma_2^{-2})^{-1/2}$.

---

> > > ### Comment · Reviewer_ah8R · 2023-08-21
> > > **Thanks for the response**
> > >
> > > I thank the authors for their response. My concerns and questions have been addressed.

---

### Official Review · Reviewer_upr1 · 2023-07-24

**Soundness:** 3 good
**Presentation:** 3 good
**Contribution:** 2 fair
**Rating:** 6
**Confidence:** 4

**Summary:**

This paper focuses on the concept of gradient sparsity in large embedding models during training, with a particular focus on privacy-preserving methods. The commonly used DP-SGD approach adds noise to all embedding gradients, even those that may not appear in the current batch, in order to ensure privacy during model updates. The authors propose algorithms aimed at preserving gradient sparsity while achieving privacy in the training of these models. By doing so, they aim to improve the efficiency and effectiveness of private training for large embedding models.

**Strengths:**

The paper is well-written and the presentation is clear, which is helpful to follow and understand the paper. The problem is clearly described, DP-SGD adding noise to all embedding gradients destroys the sparsity and the authors propose approaches for private training while keeping the sparsity for effectiveness. The experiments (in certain settings, see below) empirically support the claims of the paper.

**Weaknesses:**

I think the main weakness of the paper is that the approach is limited in the sense that it is helpful in more specific scenarios where the embedding layer is dominating the network. CTR prediction task might be a good example for this where the sparsity in embedding layer matters but for the language models, the reviewer is not so sure about the effectiveness of this approach. More details are in the questions.

**Questions:**

1) The authors use the RoBERTa model for downstream classification tasks from the GLUE benchmark. The model has 50k vocab size. Prior work that the authors cite shows that DP-SGD provides strong performance on large batch sizes (approx 1k-10k) and considering that each sample has (approx 128-512) tokens, do we expect to see the gradient sparsity phenomenon in DP-SGD training of large language models with these hyperparams? The paper shows embedding gradient sparsity in Ads model but I do not see any discussion around this regarding language models.

2) For large language models, transformer layers are in general the dominating part of the model. Could the authors provide examples for how/when this sparsification approach of the embedding layer be effective and help concretely in what sense?

3) Actually DP-SGD hyperparameters for language model experiments are not clearly provided. I only see the sentence that says "we fine-tune the clipping norm and report the best accuracy". On the other hand, it's known again from prior work that when set sufficiently small, clipping norm does not play major role in the performance of the model but batch size and learning rate are the critical hyperparameters.

Minor: Many Appendix B references seems to have been pointed to Appendix C.

I can see the contributions of the paper but perhaps rather in a more limited case where the embedding layer is the critical part of the model in terms of size and time bottleneck. It just seems to me that (large) language models may not really be the representative of this scenario.

**Limitations:**

The authors adequately addressed the limitations according to this reviewer (apart from the reviewer's questions above).

---

> ### Author Rebuttal · Authors · 2023-08-09
>
> We genuinely appreciate the valuable feedback provided by the reviewer and have addressed them in a point-by-point manner below. We are more than willing to engage in further discussions with the reviewers should any follow-up questions arise.
>
> ### **Q1. Lack of discussions of language models**
> > Prior work that the authors cite shows that DP-SGD provides strong performance on large batch sizes (approx 1k-10k) and considering that each sample has (approx 128-512) tokens, do we expect to see the gradient sparsity phenomenon in DP-SGD training of large language models with these hyperparams? The paper shows embedding gradient sparsity in Ads model but I do not see any discussion around this regarding language models.
>
> **A**:  Great question! The following table reports the number of unique tokens in the batch and the corresponding gradient sparsity for the RoBERTa model on the SST-2, QNLI, and QQP datasets. The table also includes the average sequence length for each dataset. The vocabulary size is 50,265.
>
> | **Batch size** | **SST-2 (avg. seq length: 14.4)** |  | **QQP (avg. seq length: 29.1)** |  | **QNLI (avg. seq length: 36.7)** |  |
> |---|---|---|---|---|---|---|
> |  | # unique tokens | Gradient sparsity  | # unique tokens | Gradient sparsity  | # unique tokens | Gradient sparsity  |
> | 16 | 150.6 | 0.003 | 209.2 | 0.004 | 342.0 | 0.007 |
> | 64 | 486.4 | 0.001 | 667.2 | 0.013 | 1142.2 | 0.023 |
> | 256 | 1,363.0 | 0.027 | 1,948.2 | 0.039 | 3,528.0 | 0.070 |
> | 1,024 | 3,637.2 | 0.072 | 5,062.8 | 0.101 | 9,054.0 | 0.180 |
> | 4,096 | 7,637.6 | 0.152 | 11,306.2 | 0.225 | 18,556.2 | 0.369 |
> | 16,384 | 11,982.8 | 0.238 | 21,605.6 | 0.429 | 30,204.6 | 0.600 |
> | 65,536 | 14,105.0 | 0.281 | 31,494.0 | 0.657 | 37,403.0 | 0.744 |
>
> From this table, even for large batch sizes (1,024-16,384), the number of unique tokens is still much smaller (~2-14x) than the overall vocabulary size.  Intuitively, this is because a significant portion of the vocabulary comprises uncommon tokens, which experience infrequent updates during training. Thus our algorithm can be applied even with large batch sizes.
>
> We also note that while very large batch sizes (batch size of >1M) seem essential to achieve strong performance for private pre-training [1], this is not the case for private fine-tuning [2] (batch size of ~2k).
>
> **References**:
>
> [1] Anil R, Ghazi B, Gupta V, Kumar R, Manurangsi P. Large-scale differentially private BERT. EMNLP 2022.
>
> [2] He J, Li X, Yu D, Zhang H, Kulkarni J, Lee YT, Backurs A, Yu N, Bian J. Exploring the limits of differentially private deep learning with group-wise clipping. ICLR 2023.
>
> ### **Q2. Applicability of the proposed algorithm on LLMs**
> > For large language models, transformer layers are in general the dominating part of the model. Could the authors provide examples for how/when this sparsification approach of the embedding layer be effective and help concretely in what sense?
>
> **A**: We thank the reviewer for raising this point. Our evaluation primarily centered around the RoBERTa model (vocab size: 50k) because it is a standard backbone. However, it is important to note that many high-vocabulary models exist, with vocabulary sizes 5-20x larger than RoBERTa (shown below). For these models, our proposed sparsification method, DP-AdaFEST, could offer even more pronounced benefits.
>
> | **Model** | **Vocabulary size** |
> |---|---|
> | XLM-R [1] | 250k |
> | VoCAP [2] | 250k ~ 500k |
> | XLM-V [3] | 250k ~ 1M |
>
> For example, the following are the results on the XLM-R model for the Cross-Lingual Natural Language Inference (XNLI) task when $\epsilon$=1.0. As shown, DP-AdaFEST is able to achieve **a gradient size reduction of >150x** at a minimal utility loss of 0.01.
>
> | **Utility loss compared to DP-SGD** | **Best gradient size reduction by DP-AdaFEST** |
> |---|---|
> | 0.001 | 19.84x |
> | 0.005 | 73.42x |
> | 0.01 | 162.13x |
>
> We would like to further clarify that our proposed algorithm is mainly designed to preserve gradient sparsity of embedding layers in a black box fashion. However, users have the flexibility to apply any desired algorithm to enhance model sparsity for other layers (e.g., by employing LoRA for the transformer blocks). We believe this modular approach allows users to tailor the sparsity optimization according to their specific requirements for different layers within the model.
>
> **References**:
>
> [1] Conneau A, Khandelwal K, Goyal N, Chaudhary V, Wenzek G, Guzmán F, Grave E, Ott M, Zettlemoyer L, Stoyanov V. Unsupervised cross-lingual representation learning at scale. ACL 2020.
>
> [2] Zheng B, Dong L, Huang S, Singhal S, Che W, Liu T, Song X, Wei F. Allocating large vocabulary capacity for cross-lingual language model pre-training. EMNLP 2021.
>
> [3] Liang D, Gonen H, Mao Y, Hou R, Goyal N, Ghazvininejad M, Zettlemoyer L, Khabsa M. Xlm-v: Overcoming the vocabulary bottleneck in multilingual masked language models. Arxiv preprint.
>
> ### **Q3. Hyperparameters for NLU tasks**
> > DP-SGD hyperparameters for language model experiments are not clearly provided… it's known again from prior work that when set sufficiently small, clipping norm does not play major role in the performance of the model but batch size and learning rate are the critical hyperparameters.
>
> **A**: Please note that Appendix C.1 of our submission contains the hyperparameter choices for NLU tasks. Specifically, to constrain the search space, we fix the batch size to 1024, and vary the learning rate in {5e-4, 1e-3, 2e-3, 5e-3}, and vary the clipping norm in {0.1, 1.0, 2.0, 5.0, 10.0}. For convenience, the table below contains the optimal selection of hyperparameters for various NLU tasks when $\epsilon$=1.
>
> |  | **SST-2** | **QQP** | **QNLI** |
> |---|---|---|---|
> | Learning rate | 1e-3 | 5e-3 | 5e-3 |
> | Clipping threshold | 10.0 | 10.0 | 10.0 |
>
> ### **Typo: Many Appendix B references seem to have been pointed to Appendix C.**
>
> **A**: We appreciate the reviewer’s careful reading, and we will fix these issues in the revision.

---

> > ### Comment · Reviewer_upr1 · 2023-08-14
> > **Response to the Authors**
> >
> > The reviewer appreciates the extra work the authors put in to answer questions of the reviewer. The first table shows that indeed in private fine-tuning with DP-SGD friendly hyperparameters, the sparsity phenomenon still shows up. That's helpful, thanks very much. Regarding my Q2, adding other sparsities like LoRA is an interesting point. Because it seems to me that with full fine-tuning, dominating BERT and GPT-family LLMs include embedding layer as a small part of the overall model so the sparsity may not really help much in the overall runtime etc. as the gradient in other layers will already dominate things. But indeed if one fine-tunes with LoRA + embedding layer, then things might change. But I am not really sure if this is common practice, i.e. if we are fine-tuning with LoRA, do we even need to add the embedding layer to the fine-tuning? Does it really provide an extra improvement?
> >
> > In any case, the reviewer increases their rating and thanks the authors for their responses.

---

> > > ### Author Response · Authors · 2023-08-17
> > > **Thank you & on whether fine-tuning word embeddings benefits LoRA**
> > >
> > > **A**: We appreciate the reviewer’s response and for increasing the score. We are glad that our additional results help address the reviewer’s concern.
> > >
> > > > But indeed if one fine-tunes with LoRA + embedding layer, then things might change. But I am not really sure if this is common practice, i.e. if we are fine-tuning with LoRA, do we even need to add the embedding layer to the fine-tuning? Does it really provide an extra improvement?
> > >
> > > We appreciate this follow-up question. Typically, full-parameter fine-tuning with embedding layers updated leads to better performance compared to (efficient) partial-parameter fine-tuning such as LoRA. For instance, in Table 3 of [1], full-parameter fine-tuning (FT) surpasses LoRA (LR) across most tasks. In addition, Table 15 of the LoRA paper [2] also shows that combining LoRA with prefix-embedding tuning (which injects trainable word embeddings) improves the performance of vanilla LoRA, which suggests that employing trainable word embeddings with LoRA could lead to potential improvements in utility.
> > >
> > >
> > > > Does it really provide an extra improvement?
> > >
> > > **Yes**. As demonstrated in Table 3 in our Appendix C.2.3, applying LoRA during fine-tuning, along with simultaneous updates to the embedding layer, offers better utility compared to using LoRA alone.
> > >
> > > Once again, we value the prompt feedback from the reviewer and remain available for further discussion if any new questions emerge.
> > >
> > > **References**
> > >
> > > [1] Ning Ding, Yujia Qin, Guang Yang, Fuchao Wei, Zonghan Yang, Yusheng Su, Shengding Hu et al. Delta tuning: A comprehensive study of parameter efficient methods for pre-trained language models. arXiv preprint.
> > >
> > > [2] Edward J. Hu, Yelong Shen, Phillip Wallis, Zeyuan Allen-Zhu, Yuanzhi Li, Shean Wang, Lu Wang, and Weizhu Chen. Lora: Low-rank adaptation of large language models. ICLR 2022.

---

### Author Rebuttal · Authors · 2023-08-09

We express our gratitude to the AC and all reviewers for their time and valuable feedback. We appreciate the acknowledgment that "the question addressed in this work is of great importance" and that "the proposed method is very intuitive/effective."

Below, we provide a summary of the comments and our corresponding responses for clarity. For more details, please refer to the individual responses to each reviewer. We are more than willing to engage in further discussions with the reviewers should any follow-up questions arise.

## **1. Our contribution and comparison with LoRA**
- **Novelty** (Reviewer [KMWx](https://openreview.net/forum?id=sqTcCXkG4P&noteId=JN1qnViLX3), Q1): We kindly clarified that to the best of our knowledge, no prior research has addressed sparse DP training of extensive embedding layers in recommender systems and language models. We welcome any references the reviewer might have on this topic for potential inclusion.
- **Comparison with LoRA** (Reviewer [KMWx](https://openreview.net/forum?id=sqTcCXkG4P&noteId=JN1qnViLX3), Q2): We detailed the limitations of applying LoRA to embedding models, including 1) limited room for enhancement, 2) challenges in harnessing its gradient size reduction, and 3) constraints on fine-tuning (unlike our pre-training adaptable methods). Furthermore, our empirical results show that LoRA underperforms in gradient size reduction compared to our proposed approaches.

## **2. Applicability of our methods**
- **For large batch sizes in language models** (Reviewer [upr1](https://openreview.net/forum?id=sqTcCXkG4P&noteId=T4kwfmPHxH), Q1): Our demonstrated sparsity pattern shows that even with substantial batch sizes (1,024-16,384), the unique token count remains much lower (~2-14x) than the total vocabulary size. This affirms the applicability of our algorithm to larger batch sizes.
- **For larger vocabulary language models** (Reviewer [upr1](https://openreview.net/forum?id=sqTcCXkG4P&noteId=T4kwfmPHxH), Q2): We showcased our method's capability to achieve greater gradient size reduction in language models with expanded vocabularies (e.g., 250k compared to 50k in our submission).
- **Compatibility with sparsity optimization in other layers** (Reviewer [upr1](https://openreview.net/forum?id=sqTcCXkG4P&noteId=T4kwfmPHxH), Q2): We clarified that our algorithm primarily preserves gradient sparsity in embedding layers in a black box manner. However, users retain flexibility to apply desired methods for enhancing model sparsity in other layers (e.g., using LoRA for transformer blocks).

## **3. Privacy Analysis** (Reviewer [ah8R](https://openreview.net/forum?id=sqTcCXkG4P&noteId=LZFIK6TFq5), Q1)
We clarified that our proposal is privacy-wise equivalent to a sub-sampled Gaussian mechanism, and provided more details for the privacy analysis.

## **4. Clarification questions**
We’ve also
- Clarified that the dynamically changing gradient sparsity patterns in our methods can be effectively leveraged by customized APIs such as  TPUEmbedding (Reviewer [KMWx](https://openreview.net/forum?id=sqTcCXkG4P&noteId=JN1qnViLX3), Q3);
- Explained our evaluation metric (Reviewer [KMWx](https://openreview.net/forum?id=sqTcCXkG4P&noteId=JN1qnViLX3), Q4).

## **5. Planned changes to the manuscript**
We will also make the following changes in the revision as suggested:
- (Reviewer [AuCL](https://openreview.net/forum?id=sqTcCXkG4P&noteId=XBm5ltP096), Q1) Change the title to "Sparsity-Preserving Differentially Private Training of Large Embedding Models", to more accurately reflect the scope of our work;
- (Reviewer [ah8R](https://openreview.net/forum?id=sqTcCXkG4P&noteId=LZFIK6TFq5), Q1) Provided a detailed privacy analysis for the proposed method;
- (Reviewers [KMWx](https://openreview.net/forum?id=sqTcCXkG4P&noteId=JN1qnViLX3) Q5 & [AuCL](https://openreview.net/forum?id=sqTcCXkG4P&noteId=XBm5ltP096), Q2) Report non-private baseline numbers for all datasets;
- (Reviewer [upr1](https://openreview.net/forum?id=sqTcCXkG4P&noteId=T4kwfmPHxH), Q3) Provide more details for the hyper-parameters for NLU tasks;
- (Reviewer [AuCL](https://openreview.net/forum?id=sqTcCXkG4P&noteId=XBm5ltP096), Q3) Add new references for improvements of vanilla DP-SGD;
- (Reviewer [upr1](https://openreview.net/forum?id=sqTcCXkG4P&noteId=T4kwfmPHxH)) Fix typos.

---

### Decision · Program_Chairs · 2023-09-21

**Decision:**

Accept (poster)

**Comment:**

The reviewers were thoughtful enough to participate in a lively discussion. The remaining negative reviewer clearly outlined their main gripes with the paper (listed below, verbatim). The other reviewers, who were borderline, didn't raise much disagreement with these critiques, but varied as to whether these should preclude acceptance or not. I used my own judgement, and believe that while these drawbacks are non-trivial, they are all addressable for a final camera ready version, without requiring additional careful checking by reviewers. They are mostly related to the writing, and toning down overclaiming. While NeurIPS does not have a formal "revision" option, given the public nature of these meta-reviews, I trust the authors will appropriately address them in the final version (which I look forward to reading in depth myself).

On the positive side, I and the reviewers agree that maintaining sparsity in embedding layers is an interesting, innovative, and important direction. It hasn't received much attention yet, and this paper highlights the problem and provides a nice solution. Given the merits of this paper, and the drawbacks being largely addressible, I judge that this paper should be accepted.

---

1. A Discrepancy between claimed and actual contributions
During the rebuttal, the authors specified the contribution of this work as: "...we specifically target the issue of diminished gradient sparsity encountered when employing DP-SGD with large embedding models due to the addition of dense noise." In the paper, they claim that: "To the best of our knowledge, this study is the first to address the technical challenges of applying DP-SGD to large embedding models." These two expressions are essentially different: the former suggests a specific solution, while the latter asserts a pioneering effort to address a general problem. The actual contribution of this work falls short of what is claimed in the paper.

2. Major revision
The authors overlooked that many previous works have taken the same perspective on increasing optimization efficiency. Moreover, the inherent sparsity of the embedding layer has been observed and utilized in earlier works, e.g., [1]. Additionally, the authors made a mistake in one of the main experiments, presenting the benefit of preserving sparsity (c.f. https://openreview.net/forum?id=sqTcCXkG4P&noteId=BmDe9HygLq). The authors have promised to add necessary discussions and update experimental results.

3. Experiments
Although preserving sparsity in the embedding layer can improve optimization efficiency, its effect might be marginal when the embedding layer is not dominant in the network architecture. The authors have only presented results relating to gradient reduction and efficiency improvement in the embedding layer, making the practical benefits unclear. They have promised to increase transparency regarding the final profit.